# Acquisition of cellular properties during alveolar formation requires differential activity and distribution of mitochondria

Kuan Zhang[1], Erica Yao[1], Biao Chen[1], Ethan Chuang[1], Julia Wong[1], Robert I Seed[2], Stephen L Nishimura[2], Paul J Wolters[3], Pao-Tien Chuang[1]*

[1]Cardiovascular Research Institute, University of California, San Francisco, United States; [2]Department of Pathology, University of California, San Francisco, United States; [3]Division of Pulmonary, Critical Care, Allergy and Sleep Medicine, Department of Medicine, University of California, San Francisco, United States

*For correspondence:
pao-tien.chuang@ucsf.edu

Competing interest: The authors declare that no competing interests exist.

**Abstract** Alveolar formation requires coordinated movement and interaction between alveolar epithelial cells, mesenchymal myofibroblasts, and endothelial cells/pericytes to produce secondary septa. These processes rely on the acquisition of distinct cellular properties to enable ligand secretion for cell-cell signaling and initiate morphogenesis through cellular contraction, cell migration, and cell shape change. In this study, we showed that mitochondrial activity and distribution play a key role in bestowing cellular functions on both alveolar epithelial cells and mesenchymal myofibroblasts for generating secondary septa to form alveoli in mice. These results suggest that mitochondrial function is tightly regulated to empower cellular machineries in a spatially specific manner. Indeed, such regulation via mitochondria is required for secretion of ligands, such as platelet-derived growth factor, from alveolar epithelial cells to influence myofibroblast proliferation and contraction/migration. Moreover, mitochondrial function enables myofibroblast contraction/migration during alveolar formation. Together, these findings yield novel mechanistic insights into how mitochondria regulate pivotal steps of alveologenesis. They highlight selective utilization of energy in cells and diverse energy demands in different cellular processes during development. Our work serves as a paradigm for studying how mitochondria control tissue patterning.

## Editor's evaluation

This paper will be of interest to the large class of scientists interested in lung development and disease. It explores the under-investigated role of mitochondrial activity and subcellular distribution for alveolar formation, by using a variety of transgenic mouse models to delete two specific mitochondrial proteins. The data support a role for mitochondria distribution and function in postnatal lung development in the mice.

## Introduction

Production of alveoli during development and following lung injury is essential for lung function (*Burri, 2006*; *Chao et al., 2016*; *Whitsett and Weaver, 2015*; *Pan et al., 2019*; *Rippa et al., 2021*). Defective alveologenesis underlies bronchopulmonary dysplasia (BPD) (*Silva et al., 2015*), and ongoing destruction of alveoli is characteristic of chronic obstructive lung disease (COPD) (*Patel et al., 2019*). COPD is a major cause of morbidity and mortality globally (*Barnes et al., 2015*; *Rodríguez-Castillo et al., 2018*). During alveolar formation, alveolar epithelial cells (type I [AT1] and type II [AT2] cells), myofibroblasts, and endothelial cells/pericytes undergo coordinated morphogenetic movement to

**eLife digest** The lungs display an intricate, tree-shaped structure which enables the complex gas exchanges required for life. The end of each tiny 'branch' hosts delicate air sacs, or alveoli, which are further divided by internal walls called septa. In mammals, this final structure is acquired during the last stage of lung development. Then, many different types of cells in the immature alveoli multiply and reach the right location to start constructing additional septa.

While the structural changes underlining alveoli maturation are well-studied, the energy requirements for that process remain poorly understood. In particular, the exact role of the mitochondria, the cellular compartments that power most life processes, is still unclear.

Zhang et al. therefore set out to map, in detail, the role of mitochondria in alveolar development. Microscope imaging revealed how mitochondria were unevenly distributed within the lung cells of newborn mice. Mitochondria accumulated around the machinery that controls protein secretion in the epithelial cells that line the air sacs, and around the contractile apparatus in the underlying cells (the 'myofibroblasts').

Genetically altering the mice to reduce mitochondrial activity or perturb mitochondrial location in these two cell types produced defective alveoli with fewer septa, but it had no effect on lung development before alveoli formation. This suggests that the formation of alveoli requires more energy than other steps of lung development. Disrupting mitochondrial activity or location also compromised how epithelial cells produced chemical signals necessary for the contraction or migration of the myofibroblasts.

Together, these results highlight the importance of tightly regulating mitochondrial activity and location during lung patterning. In the future, this insight could lay the groundwork to determine how energy requirements in various tissues shape other biological processes in health and disease.

generate secondary septa within saccules. As a result, secondary septa are comprised of a layer of alveolar epithelial cells that ensheathes a core of myofibroblasts and endothelial cells/pericytes. Secondary septa formation (or secondary septation) is the most important step during alveolar formation. Platelet-derived growth factor (PDGF) produced by alveolar epithelial cells is a key player in controlling myofibroblast proliferation and contraction/migration during alveologenesis (*Boström et al., 1996*; *Lindahl et al., 1997*). In response to PDGF signaling, the traditional model posits that myofibroblasts proliferate and migrate to the prospective site of secondary septation and secrete elastin. Myofibroblasts and endothelial cells/pericytes are subsequently incorporated with alveolar epithelial cells to form secondary septa. All of these principal components play a key role in driving secondary septa formation (*Chao et al., 2016*). Generation of alveoli increases the surface area and efficiency of gas exchange, enabling high activity in terrestrial environments. Despite the progress that has been made, our mechanistic understanding of alveologenesis remains incomplete.

Mitochondrial activity is essential for every biological process, and mitochondria provide a major source of ATP production through oxidative phosphorylation (OXPHOS) (*Labbé et al., 2014*; *Chan, 2020*). Unexpectedly, we have limited mechanistic insight into how mitochondria control cellular processes in vivo. In particular, little is known about whether certain cellular processes have a higher energy demand during alveolar formation. Many genetic and molecular tools have been developed in mice to study mitochondrial function. They offer a unique opportunity to address the central question of how mitochondria control alveologenesis at the molecular level.

Mitochondria exhibit dynamic distribution within individual cells. This process is mediated by the cytoskeletal elements that include microtubules, F-actin and intermediate filaments. For instance, *Rhot1* (*ras homolog family member 1*), which is also called *Miro1* (*Mitochondrial Rho GTPase 1*), encodes an atypical Ras GTPase and plays an essential role in mitochondrial transport (*Devine et al., 2016*). RHOT1 associates with the Milton adaptor (TRAK1/2) and motor proteins (kinesin and dynein), and tethers the adaptor/motor complex to mitochondria. This machinery facilitates transport of mitochondria via microtubules within mammalian cells. Whether regulated mitochondrial distribution is essential for lung cell function during alveologenesis is unknown.

In this study, we have demonstrated a central role of mitochondrial activity and distribution in conferring cellular properties to alveolar epithelial cells and myofibroblasts during alveolar formation.

In particular, PDGF ligand secretion from alveolar epithelial cells and motility of myofibroblasts depend on regulated activity and distribution of mitochondria. Moreover, loss of mitochondrial function does not have a uniform effect on cellular processes, indicating diverse energy demands in vivo. We also reveal regulation of mitochondrial function by mTOR complex 1 (mTORC1) (*Laplante and Sabatini, 2012*; *Land et al., 2014*) during alveolar formation and establish a connection between mitochondria and COPD/emphysema. Taken together, these findings provide new insight into how different cell types channel unique energy demands for cellular machinery into distinct cellular properties during alveolar formation.

## Results

### Mitochondria display dynamic subcellular distribution in alveolar epithelial cells and mesenchymal myofibroblasts during alveolar formation

To uncover the functional role of mitochondria during alveologenesis, we first examined the distribution of mitochondria in murine lung cells involved in alveolar formation. We used antibodies against mitochondrial components to visualize the distribution of mitochondria in lung epithelial cells and myofibroblasts. For instance, we performed immunostaining on lung sections derived from *Sox9$^{Cre/+}$*; *ROSA26$^{mTmG/+}$* mice with anti-mitochondrial pyruvate carrier 1 (MPC1) and anti-mitochondrially encoded cytochrome *c* oxidase I (MTCO1) (*Varuzhanyan et al., 2019*). In particular, anti-MPC1 serves as a general marker for mitochondria. Lung epithelial cells were labeled by GFP produced from the *ROSA26$^{mTmG}$* reporter (*Muzumdar et al., 2007*) due to selective Cre expression in SOX9$^+$ epithelial cells (*Akiyama et al., 2005*). We found that mitochondria were widely distributed in alveolar epithelial cells (distinguished by T1α and SPC for AT1 and AT2 cells, respectively) and myofibroblasts (marked by PDGF receptor A [PDGFRA] and smooth muscle actin [SMA]) (*Sun et al., 2000*; *Gouveia et al., 2017*; *Figure 1A*). This is consistent with an essential role of mitochondrial activity in proper functioning of lung cells. In addition, we observed an uneven subcellular distribution of mitochondria (*Figure 1A and B*). For instance, mitochondria were concentrated in areas adjacent to the trans-Golgi network (TGN38$^+$) in alveolar epithelial cells (especially AT1 cells) where proteins were sorted to reach their destinations through vesicles and in areas that surrounded SMA in myofibroblasts (*Figure 1A and B*). This finding suggests that localized mitochondrial distribution is required for cellular function in mammalian lungs.

### Compromised mitochondrial activity in the postnatal murine lung leads to defective alveologenesis

We first tested if mitochondrial activity is required for alveologenesis by inactivating *Tfam* (transcription factor A, mitochondria), which encodes a master regulator of mitochondrial transcription (*Bouda et al., 2019*), in the mouse lung after birth. We produced *CAGG$^{CreER/+}$*; *ROSA26$^{mTmG/+}$* (control), and *Tfam$^{f/f}$*; *CAGG$^{CreER/+}$*; *ROSA26$^{mTmG/+}$* mice. Tamoxifen was administered to neonatal mice to activate CreER and lungs were collected at postnatal (P) day 10 (*Figure 2A*). CreER expression under the *CAGG* promoter/enhancer (*Hayashi and McMahon, 2002*) was ubiquitous in lung cells, including NKX2.1$^+$ epithelial cells and PDGFRA$^+$ fibroblasts/myofibroblasts, and converted a floxed allele of *Tfam* (*Tfam$^f$*) (*Hamanaka et al., 2013*) into a null allele (*Figure 2B*). We noticed that multiple regions in the lungs of mutant mice displayed alveolar defects concomitant with an increased mean linear intercept (MLI), a measure of air space size (*Campbell and Tomkeieff, 1952*; *Escolar et al., 1994*; *Figure 2C and D*). Alveolar defects were associated with disorganized SMA (*Figure 2E*). We anticipated that *Tfam* removal led to shutdown of mitochondrial transcription and reduction of mitochondrial activity. Indeed, the relative ratio of mitochondrial DNA (mtDNA) (*Venegas and Halberg, 2012*), 16S rRNA, and mitochondrially encoded NADH dehydrogenase 1 (mtND1), to nuclear DNA (nDNA), hexokinase 2 (Hk2), was reduced in *Tfam*-deficient lungs compared to controls (*Figure 2F*). Loss of *Tfam* was accompanied by diminished immunoreactivity of MTCO1, the expression of which is controlled by *Tfam* (*Figure 2G*). Together, these results indicate that mitochondrial activity is required for alveolar formation.

### Selective reduction of mitochondrial activity in the lung epithelium disrupts alveologenesis

To investigate the function of mitochondrial activity in distinct compartments, we selectively reduce mitochondrial activity in either the lung epithelium or mesenchyme. We produced control and *Tfam$^{f/f}$*;

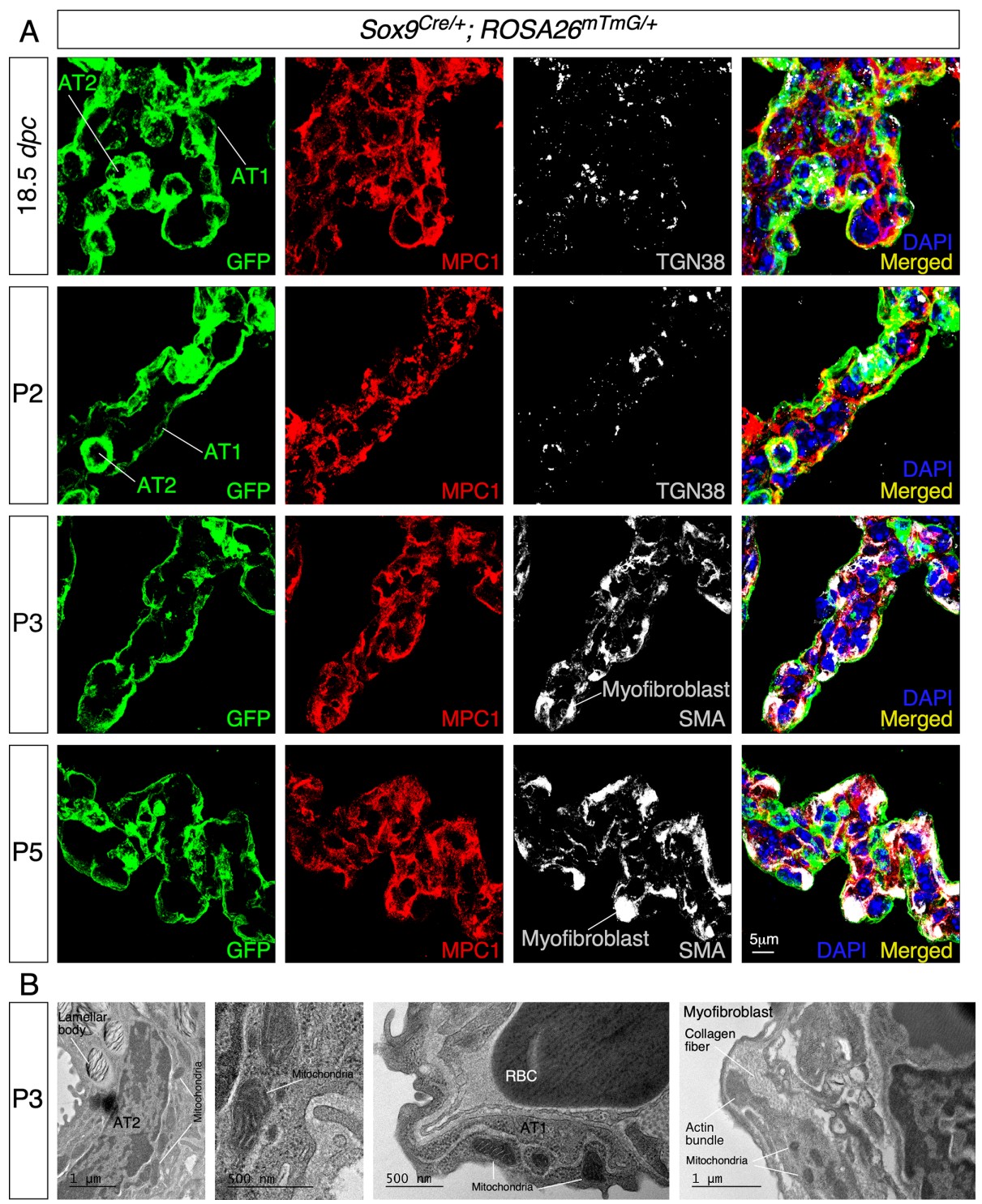

**Figure 1.** Mitochondria display subcellular concentration in alveolar epithelial cells and mesenchymal myofibroblasts of mouse lungs.
(**A**) Immunostaining of lung sections collected from *Sox9^{Cre/+}*; *ROSA26^{mTmG/+}* mice at 18.5 *days post coitus* (*dpc*) and different postnatal (P) stages as indicated. The GFP signal identified alveolar epithelial cells (alveolar type I [AT1] and alveolar type II [AT2] cells) while myofibroblasts were characterized by smooth muscle actin (SMA) expression. Moreover, mitochondria were labeled by MPC1; the trans-Golgi network was visualized by TGN38. Enhanced MPC1 signal was distributed nonuniformly in both alveolar epithelial cells and myofibroblasts. (**B**) Transmission electron micrographs of lungs collected from wild-type mice at P3. Prominent features in a given lung cell type include lamellar bodies in AT2 cells, elongated cell membrane in AT1 cells, and actin bundles and collagen fibers in myofibroblasts. RBC, red blood cell.

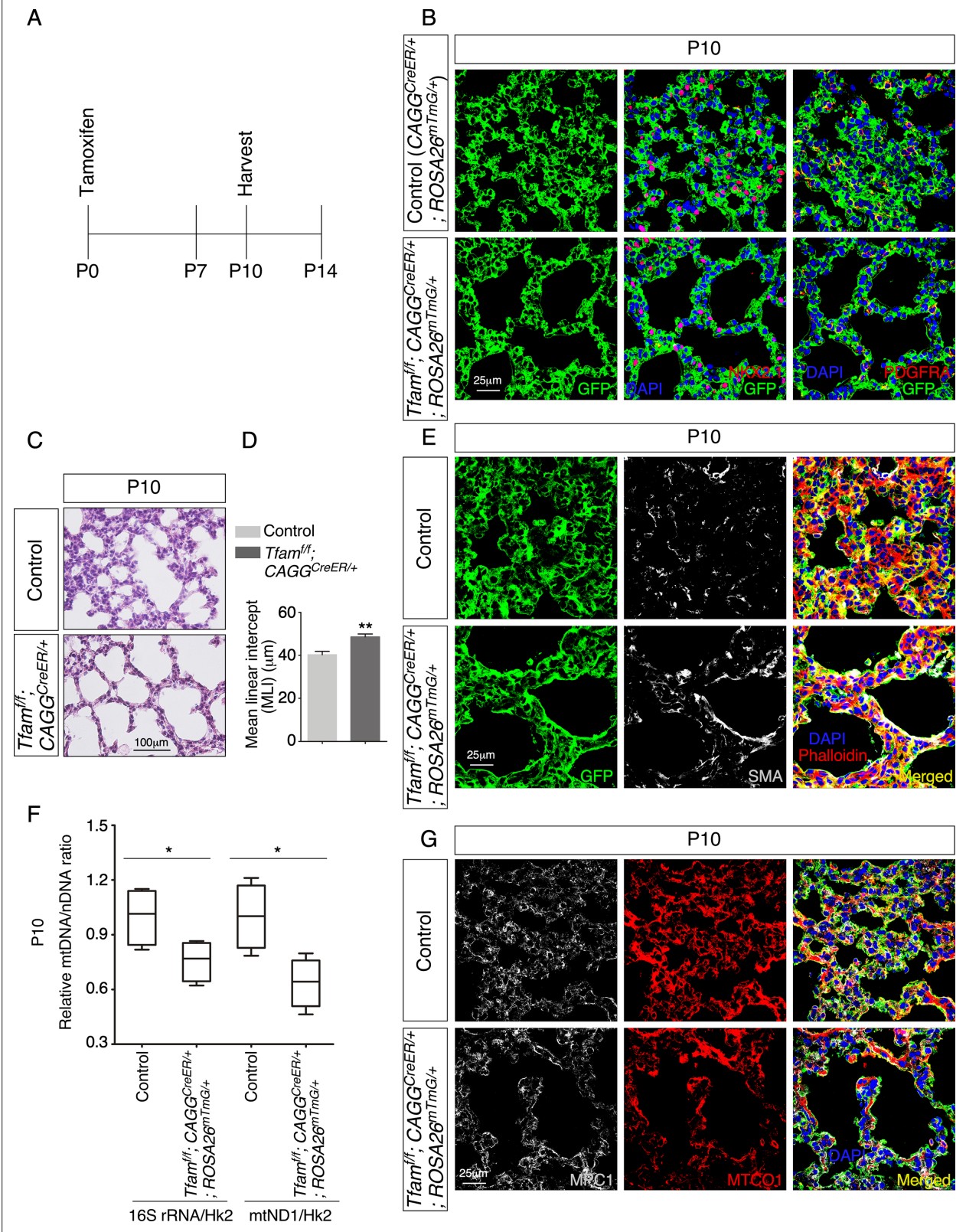

**Figure 2.** Global inactivation of *Tfam* in postnatal mice results in alveolar defects. (**A**) Schematic diagram of the time course of postnatal (P) administration of tamoxifen and harvest of mouse lungs. (**B**) Immunostaining of lungs collected from *CAGG^CreER/+; ROSA26^mTmG/+* (control) and *Tfam^f/f; CAGG^CreER/+; ROSA26^mTmG/+* mice at P10 that had received tamoxifen at P0. The GFP signal represents sites of induced CreER activity. Nuclear NKX2.1 staining marked all lung epithelial cells while PDGFRA immunoreactivity labeled mesenchymal fibroblasts/myofibroblasts. (**C**) Hematoxylin and eosin-

*Figure 2 continued on next page*

*Figure 2 continued*

stained lung sections of control and *Tfam^f/f^; CAGG^CreER/+^; ROSA26^mTmG/+^* mice at P10. Histological analysis revealed the presence of enlarged saccules and retarded development of secondary septa in the mutant lungs. (**D**) Measurement of the mean linear intercept (MLI) in control and *Tfam^f/f^; CAGG^CreER/+^; ROSA26^mTmG/+^* lungs at P10 (n = 4 for each group). The MLI was increased in *Tfam*-deficient lungs. (**E**) Immunostaining of lung sections collected from control and *Tfam^f/f^; CAGG^CreER/+^; ROSA26^mTmG/+^* mice at P10. Smooth muscle actin (SMA) expression was characteristic of myofibroblasts and phalloidin stained the actin filaments. (**F**) Quantification of the relative ratio of mitochondrial DNA (mtDNA), 16S rRNA, and mitochondrially encoded NADH dehydrogenase 1 (mtND1), to nDNA (nuclear DNA), hexokinase (Hk2), in lysates derived from control and *Tfam^f/f^; CAGG^CreER/+^; ROSA26^mTmG/+^* lungs (n = 4 for each group). (**G**) Immunostaining of lung sections collected from control and *Tfam^f/f^; CAGG^CreER/+^; ROSA26^mTmG/+^* mice at P10. MPC1 antibodies marked mitochondria; MTCO1 antibodies detected cytochrome *c* oxidase, the expression of which was controlled by *Tfam*. All values are mean ± SEM. *p<0.05; **p<0.01 (unpaired Student's *t*-test).

The online version of this article includes the following source data for figure 2:

**Source data 1.** Mean linear intercept and relative mitochondrial DNA (mtDNA)/nuclear DNA (nDNA) ratio.

*Sox9^Cre/+^* mice to establish a platform for mechanistic studies on mitochondrial activity in lung epithelial cells during alveolar formation. The *Sox9^Cre^* mouse line (**Akiyama et al., 2005**) is highly efficient in removing sequences flanked by loxP sites in the distal lung epithelium. *Sox9-Cre* is active at or later than 11.5 *days post coitus* (*dpc*) and converted *Tfam^f^* into a null allele and compromised mitochondrial activity (**Figure 3—figure supplement 1A**). *Tfam^f/f^; Sox9^Cre/+^* mice were born at the expected Mendelian frequency and could not be distinguished from their wild-type littermates by their outer appearance at birth. Moreover, histological analysis revealed no difference between control and mutant lungs prior to P5, confirming that branching morphogenesis and saccule formation were unaffected by inactivating *Tfam* in SOX9^+^ cells (**Figure 3—figure supplement 2A**). In addition, differentiation of alveolar type I and type II cells proceeded normally in *Tfam^f/f^; Sox9^Cre/+^* lungs (**Figure 3—figure supplement 2B**). These results highlight a difference in dependence on mitochondrial activity in distinct cellular processes during development. To uncover the cellular processes that are highly dependent on mitochondrial activity, we investigated alveolar formation in control and *Tfam^f/f^; Sox9^Cre/+^* lungs.

After P5, *Tfam^f/f^; Sox9^Cre/+^* mice could be discerned by their slightly reduced body size in comparison with the littermate controls. Histological analysis of *Tfam^f/f^; Sox9^Cre/+^* lungs at various postnatal stages revealed defects in secondary septa formation with an increased MLI (**Figure 3A and B**) and reduced primary septal thickness (**Figure 3C**). In this setting, primary septal thickness (P2–P7) appears to be an earlier and more sensitive indicator than MLI in detecting defects in secondary septation. No apparent difference in cell death was noted between control and *Tfam^f/f^; Sox9^Cre/+^* lungs (**Figure 3—figure supplement 3A**), suggesting that mitochondria-mediated apoptosis was not activated. Moreover, lysates from *Tfam^f/f^; Sox9^Cre/+^* lungs displayed a reduction in mtDNA/nDNA ratio (**Figure 3D**), mitochondrial complex I activity (**Figure 3E**) and ATP production (**Figure 3F**). By contrast, loss of epithelial *Tfam* did not affect the major regulators of mitochondrial fusion and fission such as OPA1 processing and DRP1 phosphorylation (**Chan, 2020**; **Figure 3—figure supplement 4**). Together, these findings are consistent with reduced mitochondrial activity in *Tfam^f/f^; Sox9^Cre/+^* lungs and reveal a critical role of mitochondrial activity in lung epithelial cells during alveologenesis. We noted that removal of *Tfam* in T cells by *Foxp3-Cre* (**Rubtsov et al., 2008**) or *Cd4-Cre* (**Lee et al., 2001**) and in macrophages by activated *Cx3cr1-CreER* (**Yona et al., 2013**) did not exhibit alveolar defects (**Figure 3—figure supplement 5**; **Fu et al., 2019**; **Gao et al., 2022**). Histological analysis revealed no difference between control and *Tfam^f/f^; Foxp3^Cre/+^* and *Tfam^f/f^; Cd4^Cre/+^* lungs, and between control and *Tfam^f/f^; Cx3cr1^CreER/+^* lungs that had received tamoxifen (**Figure 3—figure supplement 5**). This suggests that structural components of the secondary septa are more susceptible to reduced mitochondrial function.

## Disruption of mitochondrial distribution in the lung epithelium disturbs alveologenesis

As described above, mitochondria display dynamic distribution in lung cells, raising the possibility that proper subcellular distribution of mitochondria is vital for cellular function during alveolar formation. To test this hypothesis, we generated control and *Rhot1^f/f^; Sox9^Cre/+^* mice. *Sox9^Cre^* converted a floxed allele of *Rhot1* (*Rhot1^f^*) (**Nguyen et al., 2014**) to a null allele in SOX9^+^ alveolar epithelial cells. Loss of *Rhot1* is expected to perturb normal subcellular distribution of mitochondria. *Rhot1^f/f^; Sox9^Cre/+^* mice were born at the expected Mendelian frequency and cannot be distinguished from their wild-type littermates by their outer appearance or activity at birth. Similarly, no difference between control and

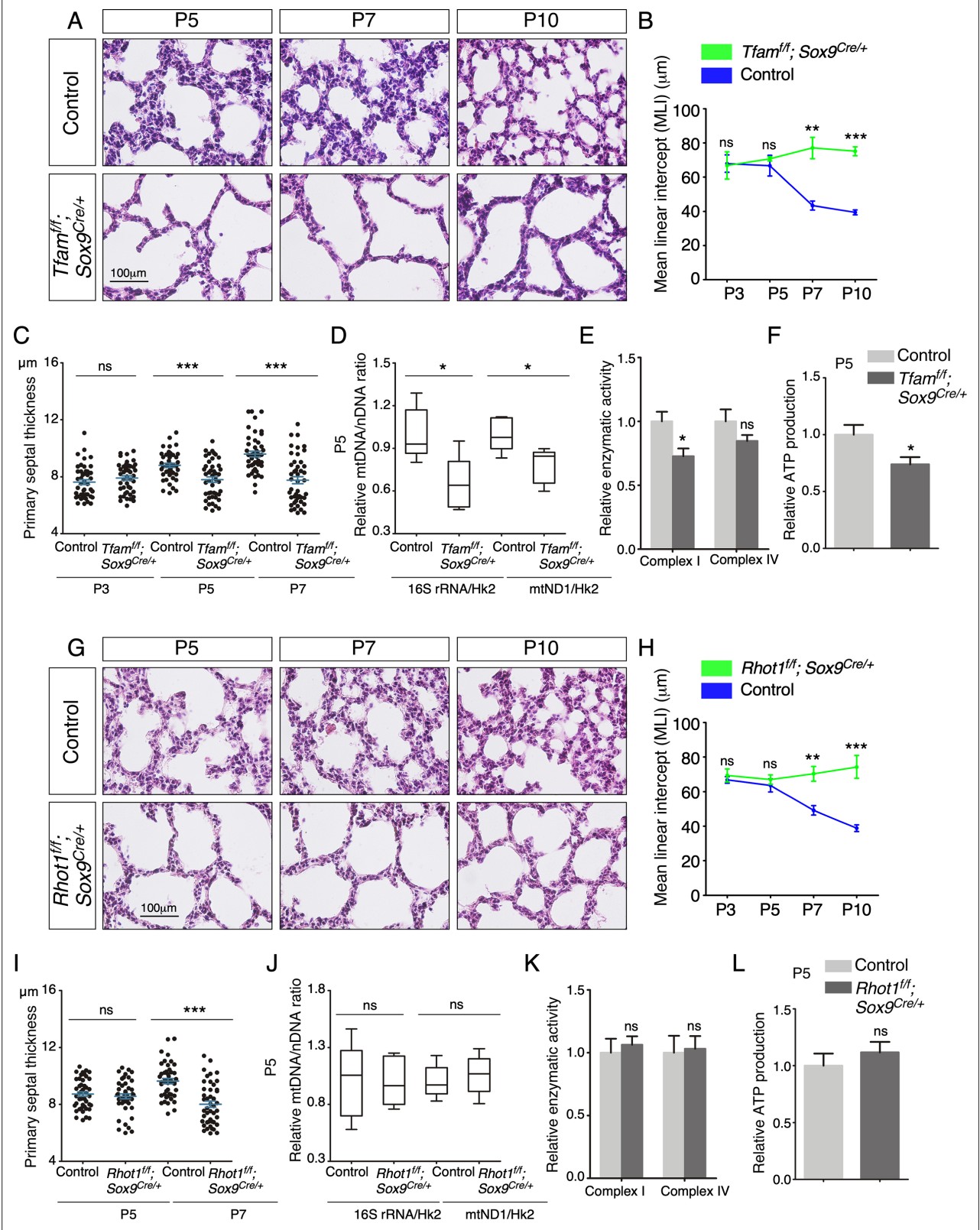

**Figure 3.** Elimination of *Tfam* or *Rhot1* in the epithelium of mouse lungs disrupts alveolar formation. (**A**) Hematoxylin and eosin-stained lung sections of control and *Tfam^f/f; Sox9^Cre/+* mice at different postnatal (P) stages as indicated. Histological analysis revealed the presence of enlarged saccules and failure in secondary septation in the mutant lungs. (**B**) Measurement of the mean linear intercept (MLI) in control and *Tfam^f/f; Sox9^Cre/+* lungs at P3–P10 (n = 3 for each group). The MLI was increased in *Tfam*-deficient lungs. (**C**) Measurement of the primary septal thickness in control and *Tfam^f/f; Sox9^Cre/+*

*Figure 3 continued on next page*

*Figure 3 continued*

lungs at P3–P7 (n = 3 for each group). (**D**) Quantification of the relative ratio of mitochondrial DNA (mtDNA), 16S rRNA, and mitochondrially encoded NADH dehydrogenase 1 (mtND1), to nuclear DNA (nDNA), hexokinase 2 (Hk2), in lysates derived from control and *Tfam^{f/f}*; *Sox9^{Cre/+}* lungs at P5 (n = 5 for each group). (**E**) Quantification of the relative enzymatic activity of mitochondrial complex I and complex IV in control and *Tfam^{f/f}*; *Sox9^{Cre/+}* lungs at P5 (n = 5 for each group). (**F**) Measurement of relative ATP production in control and *Tfam^{f/f}*; *Sox9^{Cre/+}* lungs at P5 (n = 5 for each group). (**G**) Hematoxylin and eosin-stained lung sections of control and *Rhot1^{f/f}*; *Sox9^{Cre/+}* mice at different postnatal stages as indicated. Histological analysis detected enlarged saccules and lack of secondary septa in the mutant lungs. (**H**) Measurement of the MLI in control and *Rhot1^{f/f}*; *Sox9^{Cre/+}* lungs at P3–P10 (n = 3 for each group). The MLI was increased in *Rhot1*-deficient lungs. (**I**) Measurement of the primary septal thickness in control and *Rhot1^{f/f}*; *Sox9^{Cre/+}* lungs at P5– P7 (n = 3 for each group). (**J**) Quantification of the relative ratio of mtDNA, 16S rRNA, and mtND1, to nDNA, Hk2, in lysates derived from control and *Rhot1^{f/f}*; *Sox9^{Cre/+}* lungs at P5 (n = 5 for each group). (**K**) Quantification of the relative enzymatic activity of mitochondrial complex I and complex IV in control and *Rhot1^{f/f}*; *Sox9^{Cre/+}* lungs at P5 (n = 5 for each group). (**L**) Measurement of relative ATP production in control and *Rhot1^{f/f}*; *Sox9^{Cre/+}* lungs at P5 (n = 5 for each group). All values are mean ± SEM. **p<0.01; ***p<0.001; ns, not significant (unpaired Student's *t*-test).

The online version of this article includes the following source data and figure supplement(s) for figure 3:

**Source data 1.** Mean linear intercept, primary septal thickness, relative mitochondrial DNA (mtDNA)/nuclear DNA (nDNA) ratio, relative enzymatic activity, and relative ATP production.

**Figure supplement 1.** Loss of *Tfam* or *Rhot1* disrupts mitochondrial activity and distribution, respectively.

**Figure supplement 2.** Elimination of *Tfam* or *Rhot1* in the epithelium of mouse lungs does not perturb saccule formation or cell-type specification.

**Figure supplement 3.** Loss of epithelial *Tfam* or *Rhot1* does not lead to cell death in the lungs.

**Figure supplement 4.** Loss of epithelial *Tfam* does not affect the regulators of mitochondrial fusion and fission.

**Figure supplement 5.** Loss of *Tfam* in T cells by *Foxp3–Cre* or *Cd4-Cre* and in macrophages by activated *Cx3cr1-CreER* does not perturb alveolar formation.

mutant lungs prior to P5 was detected by histological analysis (*Figure 3—figure supplement 2C and D*). After P5, *Rhot1^{f/f}*; *Sox9^{Cre/+}* mice displayed defects in secondary septation (*Figure 3G*) with an increased MLI (*Figure 3H*) and reduced primary septal thickness (*Figure 3I*). Loss of epithelial *Rhot1* did not induce cell death (*Figure 3—figure supplement 3B*). The alveolar phenotypes could first appear anywhere between P5 and P12 (*Figure 3G–I*). As expected, mitochondrial activity was unperturbed by disrupting epithelial *Rhot1*. No changes in mtDNA/nDNA ratio, mitochondrial complex I activity, or ATP production were observed in lysates from *Rhot1^{f/f}*; *Sox9^{Cre/+}* lungs (*Figure 3J–L*). These results support the notion that localized mitochondrial distribution plays a functional role in alveolar formation. We noticed that the alveolar defects in *Rhot1^{f/f}*; *Sox9^{Cre/+}* lungs were less severe than those in *Tfam^{f/f}*; *Sox9^{Cre/+}* lungs. This is likely due to the fact that only the distribution and not the activity of mitochondria was perturbed in *Rhot1^{f/f}*; *Sox9^{Cre/+}* lungs.

## PDGF signal reception is perturbed and the number of mesenchymal myofibroblasts is reduced in the absence of proper mitochondrial activity or distribution in the lung epithelium

We examined various lung cell types in *Tfam^{f/f}*; *Sox9^{Cre/+}* lungs to explore the molecular basis of their alveolar phenotypes. Interestingly, the number of fibroblasts/myofibroblasts marked by PDGFRA was reduced in the absence of epithelial *Tfam* (*Figure 4A and D*). Likewise, we found that the number of fibroblasts/myofibroblasts was reduced in *Rhot1^{f/f}*; *Sox9^{Cre/+}* lungs where epithelial *Rhot1* was lost (*Figure 4G and I*). A diminished population of fibroblasts/myofibroblasts in *Tfam^{f/f}*; *Sox9^{Cre/+}* and *Rhot1^{f/f}*; *Sox9^{Cre/+}* lungs prompted us to investigate whether PDGF signaling was disrupted.

Phosphorylation of PDGFRA (p-PDGFRA), indicative of PDGF signaling, was reduced in *Tfam^{f/f}*; *Sox9^{Cre/+}* or *Rhot1^{f/f}*; *Sox9^{Cre/+}* lungs (*Figure 4A and G*). This observation suggests that PDGF signal reception by fibroblasts/myofibroblasts was impaired in *Tfam^{f/f}*; *Sox9^{Cre/+}* or *Rhot1^{f/f}*; *Sox9^{Cre/+}* lungs. Defective PDGF signal reception in fibroblasts/myofibroblasts could be due to lack of PDGF production, trafficking, or release.

We found that production of the PDGF ligand (PDGFA) in alveolar epithelial cells was unaffected in *Tfam^{f/f}*; *Sox9^{Cre/+}* or *Rhot1^{f/f}*; *Sox9^{Cre/+}* lungs by qPCR analysis (*Figure 4E and J*). To substantiate this model, we utilized a PDGF reporter mouse line (*Pdgfa^{ex4COIN}*) (*Andrae et al., 2014*) that faithfully recapitulates the spatial and temporal expression of *Pdgfa*. Of note, no reliable PDGF antibody is available to detect PDGF in lungs or other tissues (*Gouveia et al., 2017*; *Andrae et al., 2014*). We generated *Pdgfa^{ex4COIN/+}*; *Sox9^{Cre/+}* (control) and *Tfam^{f/f}*; *Pdgfa^{ex4COIN/+}*; *Sox9^{Cre/+}* mice. Cre recombinase

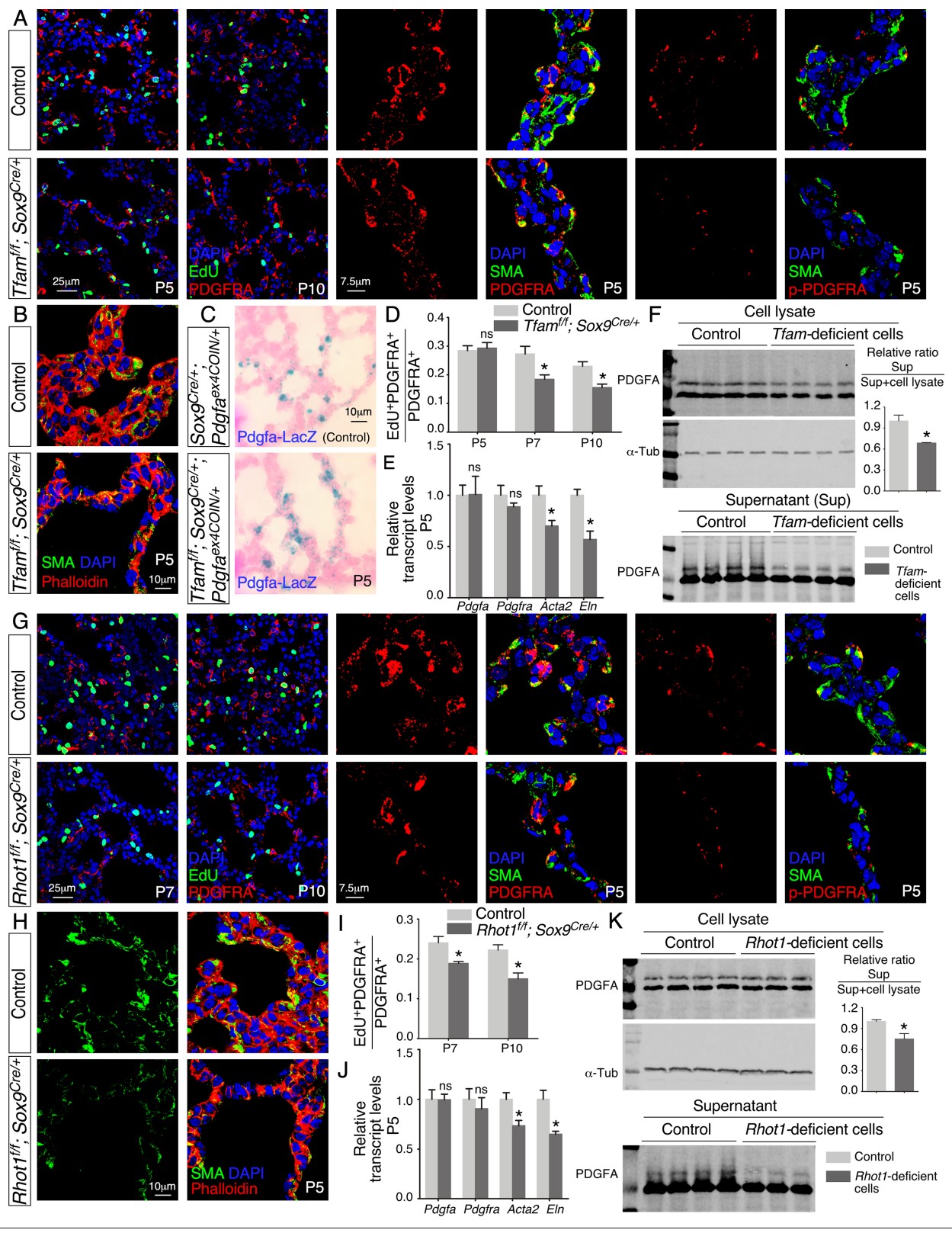

**Figure 4.** Loss of epithelial *Tfam* or *Rhot1* compromises PDGF release. (**A**) Immunostaining of lungs collected from control and *Tfam^f/f; Sox9^Cre/+* mice at postnatal (P) day 5 or 10, some of which were injected with EdU as indicated. (**B**) Immunostaining of lungs collected from control and *Tfam^f/f; Sox9^Cre/+* mice at P5. (**C**) LacZ staining (blue) of lung sections collected from *Sox9^Cre/+; Pdgfa^ex4COIN/+* (control) and *Tfam^f/f; Sox9^Cre/+; Pdgfa^ex4COIN/+* mice. The slides were counterstained with eosin (red). No difference in the intensity of LacZ staining in the lung was noted between these two mouse lines.

*Figure 4 continued on next page*

*Figure 4 continued*

(**D**) Quantification of fibroblast/myofibroblast proliferation in control and *Tfam*$^{f/f}$; *Sox9*$^{Cre/+}$ lungs at P5, P7, and P10 (n = 3 for each group). The rate of fibroblast/myofibroblast proliferation was calculated as the ratio of the number of EdU$^+$ fibroblasts/myofibroblasts (EdU$^+$ PDGFRA$^+$) to the number of fibroblasts/myofibroblasts (PDGFRA$^+$). The percentage of proliferating fibroblasts/myofibroblasts was reduced in *Tfam*$^{f/f}$; *Sox9*$^{Cre/+}$ compared to controls at P7 and P10. (**E**) qPCR analysis of gene expression in control and *Tfam*$^{f/f}$; *Sox9*$^{Cre/+}$ lungs at P5 (n = 3 for each group). While no difference in expression levels was noted for *Pdgfa* and *Pdgfra* between control and *Tfam*$^{f/f}$; *Sox9*$^{Cre/+}$ lungs, the expression levels of *Acta2* (smooth muscle actin [SMA]) and *Eln* (elastin) were significantly reduced in the absence of *Tfam*. (**F**) Western blot analysis of cell lysates and supernatants from control and *Tfam*-deficient cells (n = 4 for each group) lentivirally transduced with PDGFA-expressing constructs. The amount of PDGFA released into the media was reduced in *Tfam*-deficient cells compared to controls. α-Tubulin served as a loading control. (**G**) Immunostaining of lungs collected from control and *Rhot1*$^{f/f}$; *Sox9*$^{Cre/+}$ mice at P5, P7, or P10, some of which were injected with EdU as indicated. (**H**) Immunostaining of lungs collected from control and *Rhot1*$^{f/f}$; *Sox9*$^{Cre/+}$ mice at P5. (**I**) Quantification of fibroblast/myofibroblast proliferation in control and *Rhot1*$^{f/f}$; *Sox9*$^{Cre/+}$ lungs at P7 and P10 (n = 3 for each group). The percentage of proliferating fibroblasts/myofibroblasts was reduced in *Rhot1*$^{f/f}$; *Sox9*$^{Cre/+}$ compared to controls at P7 and P10. (**J**) qPCR analysis of gene expression in control and *Rhot1*$^{f/f}$; *Sox9*$^{Cre/+}$ lungs at P5 (n = 3 for each group). The expression levels of *Pdgfa* and *Pdgfra* were unaltered between control and *Rhot1*$^{f/f}$; *Sox9*$^{Cre/+}$ lungs; the expression levels of *Acta2* and *Eln* were significantly reduced in the absence of *Rhot1*. (**K**) Western blot analysis of cell lysates and supernatants from control and *Rhot1*-deficient cells (n = 4 for each group) lentivirally transduced with PDGFA-expressing constructs. The amount of PDGFA released into the media was reduced in *Rhot1*-deficient cells compared to controls. α-Tubulin served as a loading control. All values are mean ± SEM. *p<0.05; ns, not significant (unpaired Student's *t*-test).

The online version of this article includes the following source data and figure supplement(s) for figure 4:

**Source data 1.** EdU quantification, relative transcript levels, and quantification of PDGFA secretion.

**Figure supplement 1.** Controls for LacZ staining and lentiviral transduction.

**Figure supplement 1—source data 1.** Efficiency of lentiviral transduction.

activated *β-galactosidase* (*lacZ*) expression in *Pdgfa*-expressing cells from the *Pdgfa*$^{ex4COIN}$ allele. We found that *LacZ* expression in *Pdgfa*-expressing cells (i.e., alveolar epithelial cells) displayed a similar pattern and intensity between control and *Tfam*-deficient lungs (*Figure 4C*, *Figure 4—figure supplement 1A*). Together, these results pointed to disrupted PDGF trafficking or release. This defect would subsequently disturb signal reception in mesenchymal fibroblasts/myofibroblasts of *Tfam*$^{f/f}$; *Sox9*$^{Cre/+}$ and *Rhot1*$^{f/f}$; *Sox9*$^{Cre/+}$ lungs.

## PDGF secretion from lung cells is diminished without proper mitochondrial activity or distribution

Our model posits that secretion of PDGF ligand from *Tfam*- and *Rhot1*-deficient alveolar epithelial cells is compromised. To test this idea, we derived *Tfam*- and *Rhot1*-deficient cells from *Tfam*$^{f/f}$; *Pdgfra*$^{Cre/+}$ and *Rhot1*$^{f/f}$; *Pdgfra*$^{Cre/+}$ lungs (see below), respectively. We transduced control and *Tfam*- and *Rhot1*-deficient cells with lentiviruses that produced epitope-tagged PDGF (*Figure 4—figure supplement 1B*). Using this assay, we determined the amount of PDGF released from control and *Tfam*- and *Rhot1*-deficient cells (*Figure 4F and K*). PDGF levels in the conditioned media derived from *Tfam*- or *Rhot1*-deficient cells were reduced compared to controls (*Figure 4F and K*). These findings support a model in which loss of mitochondrial activity or distribution results in a failure of vesicular transport and PDGF release from alveolar epithelial cells. We surmise that these defects are in part due to an incapacitated actomyosin cytoskeleton caused by reduced mitochondrial function.

## Selective reduction of mitochondrial activity or distribution in fibroblasts/myofibroblasts compromises alveologenesis

We then investigated the functional requirement of mitochondria in the lung mesenchyme. To this end, we produced control and *Tfam*$^{f/f}$; *Pdgfra*$^{Cre/+}$ (*Roesch et al., 2008*) mice for *Tfam* inactivation in lung fibroblasts/myofibroblasts. It was reported that ~95% of lineaged cells (PDGFRA$^+$) are myofibroblasts (*Li et al., 2018*). Nevertheless, we have adopted PDGFRA$^+$ fibroblasts/myofibroblasts throughout this study for accuracy. Cre expression in PDGFRA$^+$ fibroblasts/myofibroblasts eliminated *Tfam* and reduced mitochondrial activity. Indeed, the expression of MTCO1, a transcriptional target of *Tfam*, was decreased in lung fibroblasts/myofibroblasts derived from *Tfam*$^{f/f}$; *Pdgfra*$^{Cre/+}$ mice (*Figure 3—figure supplement 1B*). Prior to P3, *Tfam*$^{f/f}$; *Pdgfra*$^{Cre/+}$ mice could not be distinguished from their wild-type littermates by appearance, activity, and morphological and immunohistochemical analysis (*Figure 5—figure supplement 1A and B*). This suggests that loss of *Tfam* in the lung mesenchyme did not affect branching morphogenesis or saccule formation (*Metzger et al., 2008*). This permitted

us to assess the contribution of mesenchymal mitochondria to alveolar formation. Histological analysis of *Tfam^{f/f}; Pdgfra^{Cre/+}* lungs at various postnatal stages prior to P5 revealed defective secondary septa formation with an increased MLI (*Figure 5A and B*) and reduced primary septal thickness (*Figure 5C*). Loss of mesenchymal *Tfam* did not induce cell death (*Figure 5—figure supplement 2A*). Moreover, lysates from *Tfam^{f/f}; Pdgfra^{Cre/+}* lungs displayed a reduction in mtDNA/nDNA ratio (*Figure 5D*), mitochondrial complex I/IV activity (*Figure 5E*), and ATP production (*Figure 5F*). All of them are consistent with reduced mitochondrial activity in *Tfam^{f/f}; Pdgfra^{Cre/+}* lungs. Most *Tfam^{f/f}; Pdgfra^{Cre/+}* mice died before P30.

We also generated control and *Tfam^{f/f}; Twist2* (*Dermo1*)^{Cre/+} mice, in which *Tfam* was eliminated by mesenchymal *Twist2-Cre* (*Yu et al., 2003*). *Tfam^{f/f}; Twist2^{Cre/+}* mice exhibited alveolar defects (*Figure 5—figure supplement 3A and B*) similar to those in *Tfam^{f/f}; Pdgfra^{Cre/+}* mice, further supporting a central role of mitochondrial activity in the lung mesenchyme during alveologenesis.

Of note, we bred *Lrpprc^{f/f}; Pdgfra^{Cre/+}* mice as an alternative means to disrupt mitochondrial activity. *Lrpprc* (*Leucine-rich PPR motif-containing*) (*Ruzzenente et al., 2012*) is required for mitochondrial translation. Similarly, *Pdgfra^{Cre}* converted a floxed allele of *Lrpprc* (*Lrpprc^{f}*) into a null allele. *Lrpprc* affects different aspects of mitochondrial activity and serves the purpose of confirming our findings using *Tfam*. We found that *Lrpprc^{f/f}; Pdgfra^{Cre/+}* mice developed alveolar defects (*Figure 5—figure supplement 3C and D*), albeit the phenotypes were less severe than those in *Tfam^{f/f}; Pdgfra^{Cre/+}* mice. This was likely due to the presence of residual proteins after removal of *Lrpprc*. Together, these studies establish a crucial role of mitochondrial activity in fibroblasts/myofibroblasts for alveologenesis.

We went on to determine whether proper subcellular distribution of mitochondria in fibroblasts/myofibroblasts is necessary for their function during alveolar formation. We generated control and *Rhot1^{f/f}; Pdgfra^{Cre/+}* mice (*Figure 5—figure supplement 1C and D*). In this case, the subcellular distribution of mitochondria is expected to be perturbed in mesenchymal fibroblasts/myofibroblasts as indicated by loss of proper MTCO1 distribution in fibroblasts/myofibroblasts derived from lungs of *Rhot1^{f/f}; Pdgfra^{Cre/+}* mice (*Figure 3—figure supplement 1C*). *Rhot1^{f/f}; Pdgfra^{Cre/+}* mice exhibited alveolar defects (*Figure 5G–I, Figure 5—figure supplement 2B*), milder than those in *Tfam^{f/f}; Pdgfra^{Cre/+}* lungs. Mitochondrial activity was unperturbed by disrupting mesenchymal *Rhot1* (*Figure 5J–L*).

We discovered that fibroblast/myofibroblast proliferation was reduced in *Tfam^{f/f}; Pdgfra^{Cre/+}* or *Rhot1^{f/f}; Pdgfra^{Cre/+}* lungs compared to controls (*Figure 6A, B, E and F*), suggesting defective PDGF signal reception. This may be related to a failure in PDGFR trafficking when mitochondrial function is impaired. As a result, the pool of PDGFRA⁺ myofibroblasts was decreased with a concomitant reduction in SMA (encoded by *Acta2*) and/or elastin (*Figure 6C and G*), contributing to alveolar defects. In summary, these findings suggest that proper activity and distribution of mitochondria in alveolar fibroblasts/myofibroblasts are critical to generating a sufficient number of fibroblasts/myofibroblasts for secondary septation.

## Contraction/migration of fibroblasts/myofibroblasts is reduced without proper mitochondrial activity or distribution, which is associated with a disrupted cytoskeleton

We speculate that myofibroblasts deficient in *Tfam* are defective in their ability to migrate to the prospective site of secondary septation. To test this idea, we isolated fibroblasts/myofibroblasts from control and *Tfam^{f/f}; Pdgfra^{Cre/+}* lungs. *Tfam*-deficient fibroblasts/myofibroblasts displayed short tubular and fragmented mitochondria (*Figure 6—figure supplement 1A and B*). Fibroblasts/myofibroblasts were seeded onto the migration chamber for wound healing assays (*Zhang et al., 2020*), in which the rate of fibroblasts/myofibroblasts migration into the cell-free area was measured. While control fibroblasts/myofibroblasts occupied the cell-free area after 36–48 hr, only scant *Tfam*-deficient fibroblasts/myofibroblasts were detected in the cell-free area (*Figure 6D*). Introduction of TFAM into *Tfam*-deficient fibroblasts/myofibroblasts rescued their migratory defects (*Figure 6—figure supplement 2A, B, and E*). This result indicates that mobility of fibroblasts/myofibroblasts was compromised due to loss of mitochondrial activity in these cells. We conjecture that the migration defect was in part due to an incapacitated actomyosin cytoskeleton without mitochondrial activity. Consistent with this idea, organization of the cytoskeleton (labeled by phalloidin) was perturbed in *Tfam^{f/f}; Pdgfra^{Cre/+}* lungs in comparison with controls (*Figure 6A*).

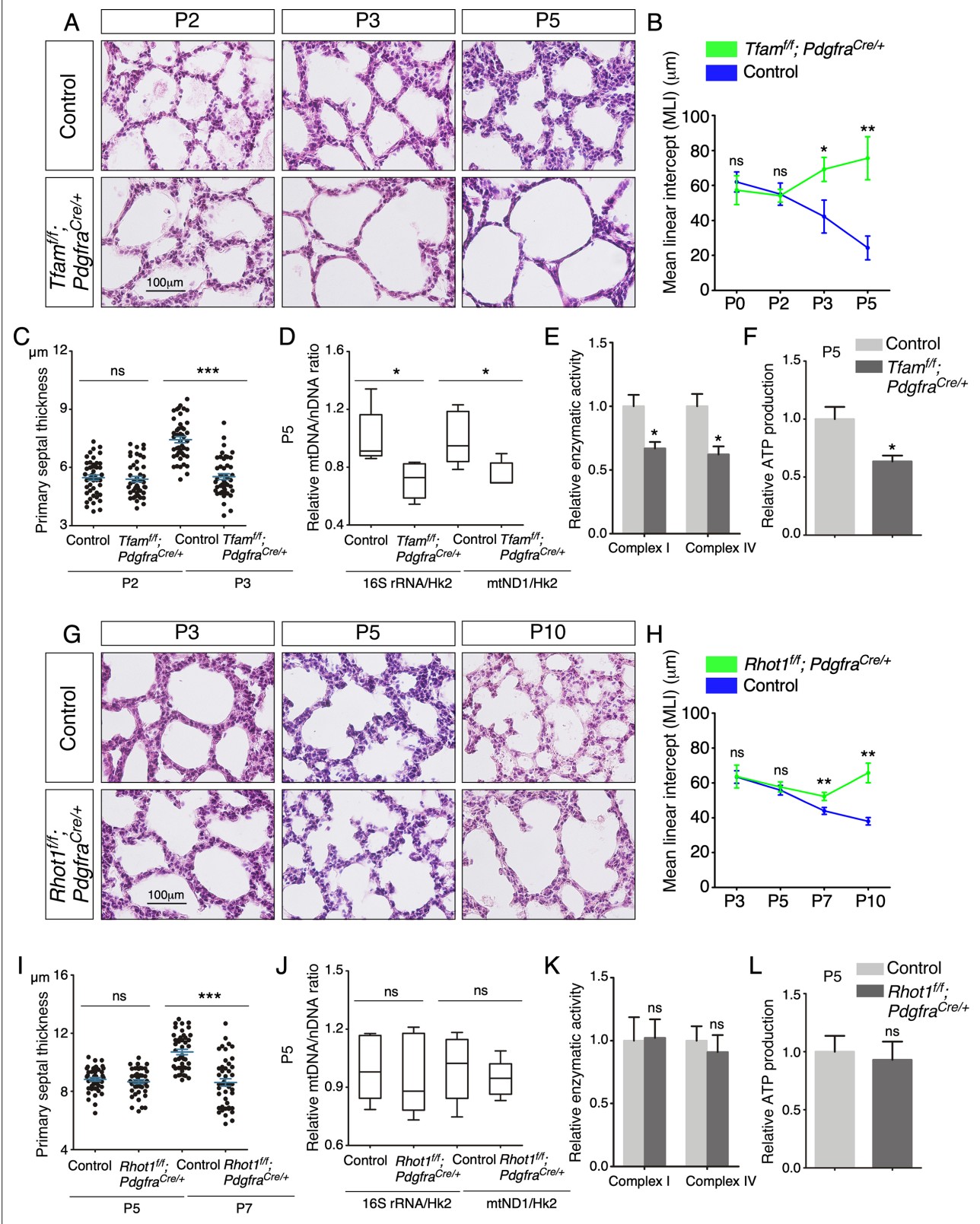

**Figure 5.** Elimination of *Tfam* or *Rhot1* in mouse lung fibroblasts/myofibroblasts impairs alveolar formation. (**A**) Hematoxylin and eosin-stained lung sections of control and *Tfam^{f/f}; Pdgfra^{Cre/+}* mice at different postnatal (P) stages as indicated. Histological analysis revealed the presence of enlarged saccules and defective development of secondary septa in the mutant lungs. (**B**) Measurement of the mean linear intercept (MLI) in control and *Tfam^{f/f}; Pdgfra^{Cre/+}* lungs at P0–P5 (n = 3 for each group). The MLI was increased in *Tfam*-deficient lungs. (**C**) Measurement of the primary septal thickness in

*Figure 5 continued on next page*

*Figure 5 continued*

control and *Tfam^{f/f}*; *Pdgfra^{Cre/+}* lungs at P2–P3 (n = 3 for each group). (**D**) Quantification of the relative ratio of mitochondrial DNA (mtDNA), 16S rRNA, and mitochondrially encoded NADH dehydrogenase 1 (mtND1), to nuclear DNA (nDNA), hexokinase 2 (Hk2), in lysates derived from control and *Tfam^{f/f}*; *Pdgfra^{Cre/+}* lungs at P5 (n = 5 for each group). (**E**) Quantification of the relative enzymatic activity of mitochondrial complex I and complex IV in control and *Tfam^{f/f}*; *Pdgfra^{Cre/+}* lungs at P5 (n = 5 for each group). (**F**) Measurement of relative ATP production in control and *Tfam^{f/f}*; *Pdgfra^{Cre/+}* lungs at P5 (n = 5 for each group). (**G**) Hematoxylin and eosin-stained lung sections of control and *Rhot1^{f/f}*; *Pdgfra^{Cre/+}* mice at different postnatal stages as indicated. Histological analysis detected enlarged saccules and lack of secondary septa in the mutant lungs. (**H**) Measurement of the MLI in control and *Rhot1^{f/f}*; *Pdgfra^{Cre/+}* lungs at P3–P10 (n = 3 for each group). The MLI was increased in *Rhot1*-deficient lungs. (**I**) Measurement of the primary septal thickness in control and *Rhot1^{f/f}*; *Pdgfra^{Cre/+}* lungs at P5–P7 (n = 3 for each group). (**J**) Quantification of the relative ratio of mtDNA, 16S rRNA, and mtND1, to nDNA, Hk2, in lysates derived from control and *Rhot1^{f/f}*; *Pdgfra^{Cre/+}* lungs at P5 (n = 5 for each group). (**K**) Quantification of the relative enzymatic activity of mitochondrial complex I and complex IV in control and *Rhot1^{f/f}*; *Pdgfra^{Cre/+}* lungs at P5 (n = 5 for each group). (**L**) Measurement of relative ATP production in control and *Rhot1^{f/f}*; *Pdgfra^{Cre/+}* lungs at P5 (n = 5 for each group). All values are mean ± SEM. *p<0.05; **p<0.01; ns, not significant (unpaired Student's *t*-test).

The online version of this article includes the following source data and figure supplement(s) for figure 5:

**Source data 1.** Mean linear intercept, primary septal thickness, relative mitochondrial DNA (mtDNA)/nuclear DNA (nDNA) ratio, relative enzymatic activity, and relative ATP production.

**Figure supplement 1.** Removal of *Tfam* or *Rhot1* in mouse lung fibroblasts/myofibroblasts does not perturb saccule formation or cell-type specification.

**Figure supplement 2.** Loss of mesenchymal *Tfam* or *Rhot1* does not lead to cell death in the lungs.

**Figure supplement 3.** Inactivation of *Tfam* or *Lrpprc* in mouse lung fibroblasts/myofibroblasts leads to alveolar defects.

**Figure supplement 3—source data 1.** Mean linear intercept.

---

Similarly, fibroblasts/myofibroblasts derived from *Rhot1^{f/f}*; *Pdgfra^{Cre/+}* lungs displayed a compromised response in wound healing assays compared to controls (***Figure 6H***, ***Figure 6—figure supplement 1C and D***), which was restored by RHOT1 expression (***Figure 6—figure supplement 2C–E***). To sum up, these results affirm the role of mitochondrial activity and distribution in regulating fibroblasts/myofibroblasts contraction and/or migration during alveologenesis.

## mTOR complex 1 regulates mitochondrial function and alveologenesis

We have discovered an essential role of mitochondria in controlling alveolar formation. To dissect the signaling cascade that regulates mitochondrial function during alveologenesis, we first investigated mTOR complex 1 (mTORC1), a known regulator of mitochondrial activity and biogenesis. We generated control, *Rptor^{f/f}*; *Sox9^{Cre/+}* and *Rptor^{f/f}*; *Pdgfra^{Cre/+}* mice. *Rptor* (*Raptor*, *rapamycin-sensitive regulatory associated protein of mTOR*) (***Sengupta et al., 2010***) encodes an essential component of mTORC1, which includes RPTOR, mTOR, and several other proteins. *Rptor^{f/f}*; *Sox9^{Cre/+}* and *Rptor^{f/f}*; *Pdgfra^{Cre/+}* mice failed to survive to term, precluding the analysis of potential alveolar phenotypes.

To circumvent this problem, we produced control and *Rptor^{f/f}*; *CAGG^{CreER/+}* mice and administered tamoxifen postnatally (***Figure 7A***). Analysis of *Rptor^{f/f}*; *CAGG^{CreER/+}* mice at P10 revealed alveolar defects with an increased MLI (***Figure 7B and C***). Defective secondary septation in *Rptor^{f/f}*; *CAGG^{CreER/+}* lungs was associated with reduced SMA (***Figure 7D***). As expected, the protein levels of phosphorylated ribosomal protein S6 (p-RPS6), a downstream target of mTORC1, were decreased in lysates from *Rptor*-deficient lungs compared to controls (***Figure 7E***, ***Figure 7—figure supplement 1***). The relative ratio of mtDNA to nDNA was reduced in *Rptor*-deficient lungs (***Figure 7F***). Moreover, immunoreactivity of both MPC1 and MTCO1 was diminished in *Rptor^{f/f}*; *CAGG^{CreER/+}* lungs compared to controls (***Figure 7G***). These findings show that elimination of *Rptor* in the lung resulted in loss of mitochondria. Impaired mitochondrial function then contributed to alveolar defects. Collectively, our results suggest that mTORC1 controls alveolar formation partly through its effects on mitochondrial function.

## Mitochondrial copy number and TFAM protein levels are decreased in lungs from COPD/emphysema patients

To explore whether studies of mitochondrial function in alveolar formation in mice can shed new light on human diseases, we assessed the status of mitochondria in lung tissues of normal subjects and COPD/emphysema patients (***Figure 8A***, ***Figure 8—figure supplement 1***). We found that the copy number of mitochondria (***Venegas and Halberg, 2012***) relative to nuclear DNA was significantly reduced in COPD/emphysema patients (***Figure 8B***). In addition, TFAM protein levels were reduced

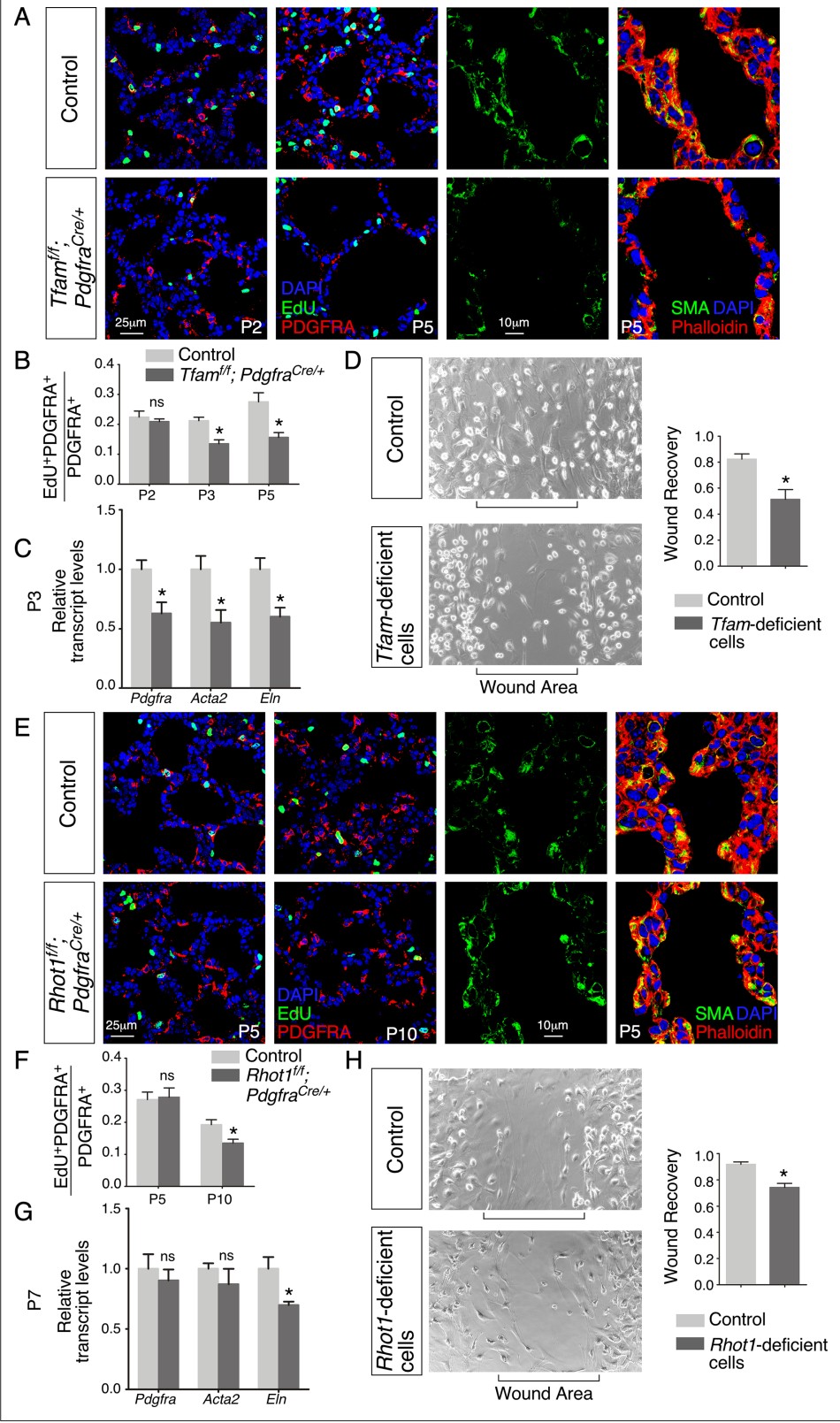

**Figure 6.** Loss of mesenchymal *Tfam* or *Rhot1* compromises fibroblast/myofibroblast migration.
(**A**) Immunostaining of lungs collected from control and *Tfam^f/f^; Pdgfra^Cre/+^* mice at postnatal (P) day 2 and 5, some of which were injected with EdU as indicated. (**B**) Quantification of fibroblast/myofibroblast proliferation in control and *Tfam^f/f^; Pdgfra^Cre/+^* lungs at P2, P3, and P5 (n = 3 for each group). The rate of fibroblast/myofibroblast

*Figure 6 continued on next page*

*Figure 6 continued*

proliferation was calculated as the ratio of the number of EdU$^+$ fibroblasts/myofibroblasts (EdU$^+$ PDGFRA$^+$) to the number of fibroblast/myofibroblasts (PDGFRA$^+$). The percentage of proliferating fibroblasts/myofibroblasts was reduced in *Tfam$^{f/f}$; Pdgfra$^{Cre/+}$* compared to controls at P3 and P5. (**C**) qPCR analysis of gene expression in control and *Tfam$^{f/f}$; Pdgfra$^{Cre/+}$* lungs at P3 (n = 3 for each group). The expression levels of *Pdgfra*, *Acta2*, and *Eln* were significantly reduced in *Tfam$^{f/f}$; Pdgfra$^{Cre/+}$* lungs compared to controls. (**D**) Wound recovery assays to assess the migratory ability of fibroblasts/myofibroblasts derived from control and *Tfam$^{f/f}$; Pdgfra$^{Cre/+}$* lungs (n = 3 for each group). Within 36–48 hr, the wound area has been populated by migrating fibroblasts/myofibroblasts derived from control lungs. By contrast, fewer fibroblasts/myofibroblasts from *Tfam$^{f/f}$; Pdgfra$^{Cre/+}$* lungs reached the wound area within the same time frame. Wound recovery by fibroblasts/myofibroblasts from control and *Tfam$^{f/f}$; Pdgfra$^{Cre/+}$* lungs was quantified. (**E**) Immunostaining of lungs collected from control and *Rhot1$^{f/f}$; Pdgfra$^{Cre/+}$* mice at P5 and P10, some of which were injected with EdU as indicated. (**F**) Quantification of fibroblasts/myofibroblasts proliferation in control and *Rhot1$^{f/f}$; Pdgfra$^{Cre/+}$* lungs at P5 and P10 (n = 3 for each group). The percentage of proliferating fibroblasts/myofibroblasts was decreased in *Rhot1$^{f/f}$; Pdgfra$^{Cre/+}$* compared to controls at P10. (**G**) qPCR analysis of gene expression in control and *Rhot1$^{f/f}$; Pdgfra$^{Cre/+}$* lungs at P7 (n = 3 for each group). The expression levels of *Eln* were significantly reduced in *Rhot1$^{f/f}$; Pdgfra$^{Cre/+}$* lungs in comparison with controls. (**H**) Wound recovery assays to assess the migratory ability of fibroblasts/myofibroblasts derived from control and *Rhot1$^{f/f}$; Pdgfra$^{Cre/+}$* lungs (n = 3 for each group). Fewer fibroblasts/myofibroblasts from *Rhot1$^{f/f}$; Pdgfra$^{Cre/+}$* lungs reached the wound area within the same time frame compared to controls. Wound recovery by fibroblasts/myofibroblasts from control and *Rhot1$^{f/f}$; Pdgfra$^{Cre/+}$* lungs was quantified. All values are mean ± SEM. *p<0.05; ns, not significant (unpaired Student's *t*-test).

The online version of this article includes the following source data and figure supplement(s) for figure 6:

**Source data 1.** EdU quantification, relative transcript levels, and quantification of wound recovery.

**Figure supplement 1.** *Tfam*- but not *Rhot1*-deficient fibroblasts display short tubular and fragmented mitochondria.

**Figure supplement 1—source data 1.** Quantification of mitochondrial morphology.

**Figure supplement 2.** The migratory defect in *Tfam*- and *Rhot1*-deficient fibroblasts/myofibroblasts is rescued by expression of TFAM and RHOT1, respectively.

**Figure supplement 2—source data 1.** Quantification of wound recovery.

in lysates from emphysema lungs compared to normal lungs (*Figure 8C*). These results established a connection between mitochondrial function and pathogenesis of COPD/emphysema. Interestingly, a disorganized cytoskeleton was noted in COPD/emphysema lungs, in which actin bundles seen in normal alveoli were sparse (*Figure 8D*). We did not detect a difference in the protein levels of either RPS6 or p-RPS6 in lysates from emphysema and normal lungs (*Figure 8E*, *Figure 8—figure supplement 2*). However, we observed heterogeneous expression of p-RPS6 in emphysema lungs. Whether regional reduction of p-RPS6 is correlated with disease progression needs future studies.

To further explore the connection between mitochondrial function and cellular properties, we conducted gain- and loss-of-function studies on *TFAM* and *RHOT1* in both human lung epithelial cells and human lung fibroblasts. *TFAM* or *RHOT1* knockdown in human lung epithelial cells impaired PDGF secretion from epithelial cells (*Figure 8F*, *Figure 8—figure supplement 3A*). This defect was rescued by introduction of mouse TFAM or RHOT1 (*Figure 8F*, *Figure 8—figure supplement 3A*). In addition, *TFAM* or *RHOT1* knockdown in human lung fibroblasts compromised cell migration of fibroblasts (*Figure 8*G, *Figure 8—figure supplement 3B*). Likewise, migration defects were restored upon expression of mouse TFAM or RHOT1 (*Figure 8*G, *Figure 8—figure supplement 3B*). Results from the experiments using human lung cells affirmed the observations obtained in mouse cells and mouse lungs. Taken together, these findings using human lungs and cells complement our mouse work and lay the foundation for further investigation into the disease mechanisms of COPD/emphysema.

## Discussion

Our studies have provided new insight into how mitochondrial function controls alveolar formation. We discovered a major role of mitochondria in conferring requisite cellular properties to both alveolar epithelial cells and myofibroblasts during alveologenesis. These findings define the molecular basis of the functional requirement of mitochondria in distinct compartments and reveal the energy demand in a given process. They also add a new layer of complexity to the interactions between the major

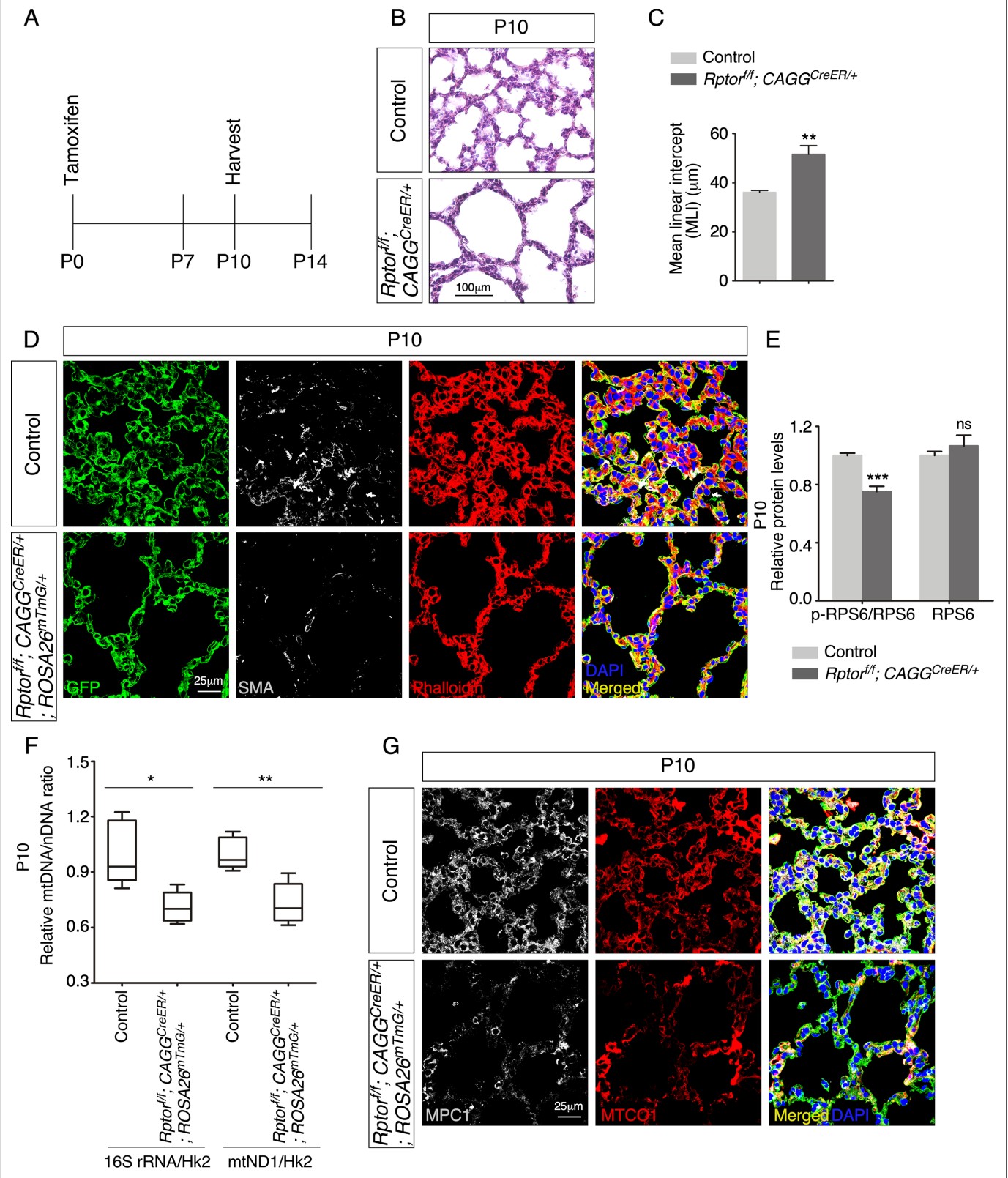

**Figure 7.** Postnatal inactivation of *Rptor* in mice results in alveolar defects. (**A**) Schematic diagram of the time course of postnatal (P) administration of tamoxifen and harvest of mouse lungs. (**B**) Hematoxylin and eosin-stained lung sections of control and *Rptor^f/f; CAGG^CreER/+* mice at P10. Histological analysis revealed the presence of enlarged saccules and retarded development of secondary septa in the mutant lungs. (**C**) Measurement of the mean linear intercept (MLI) in control and *Rptor^f/f; CAGG^CreER/+* lungs at P10 (n = 5 for each group). The MLI was increased in *Rptor*-deficient lungs.

*Figure 7 continued on next page*

Figure 7 continued

(D) Immunostaining of lung sections collected from control and *Rptor*^*f/f*^; *CAGG*^*CreER/+*^; *ROSA26*^*mTmG/+*^ mice at P10. Smooth muscle actin (SMA) expression was characteristic of myofibroblasts and phalloidin stained the actin filaments. (E). Quantification of the protein levels of RPS6 and phosphorylated RPS6 (p-RPS6) in lung lysates derived from control and *Rptor*^*f/f*^; *CAGG*^*CreER/+*^; *ROSA26*^*mTmG/+*^ lungs (n = 5 for each group). (F) Quantification of the relative ratio of mitochondrial DNA (mtDNA), 16S rRNA, and mitochondrially encoded NADH dehydrogenase 1 (mtND1), to nuclear DNA (nDNA), hexokinase 2 (Hk2), in lysates derived from control and *Rptor*^*f/f*^; *CAGG*^*CreER/+*^; *ROSA26*^*mTmG/+*^ lungs (n = 5 for each group). (G) Immunostaining of lung sections collected from control and *Rptor*^*f/f*^; *CAGG*^*CreER/+*^; *ROSA26*^*mTmG/+*^ mice at P10. MPC1 antibodies marked mitochondria; MTCO1 antibodies detected cytochrome *c* oxidase, the expression of which was controlled by *Tfam*. All values are mean ± SEM. p<0.05; **p<0.01; ***p<0.001 (unpaired Student's *t*-test).

The online version of this article includes the following source data and figure supplement(s) for figure 7:

**Source data 1.** Mean linear intercept, relative protein levels, and relative mitochondrial DNA (mtDNA)/nuclear DNA (nDNA) ratio.

**Figure supplement 1.** The ratio of p-RPS6 to RPS6 is reduced in the absence of *Rptor*.

players of secondary septa. Moreover, our work establishes the foundation for investigating the interplay between mitochondria and signaling pathways in endowing cellular properties in alveologenesis during development and following injury. We expect that this framework will be applicable to other developmental systems (*Figure 9*).

A reduction in mitochondrial function either globally or in distinct compartments in the lung results in alveolar defects. We found that not all cellular processes are perturbed to the same extent when mitochondrial activity or distribution is perturbed. For instance, while alveolar development is disrupted, saccule formation and cell-type specification are unaffected. These observations highlight the differential requirement of mitochondrial function in a given cellular process. Events that necessitate a higher demand for energy will be the first to exhibit phenotypes once mitochondrial function is compromised. It implies that the severity of phenotypes resulting from a reduction in mitochondrial function could be used to characterize the energy demand for a particular cellular process in development. This information cannot be obtained from cell-based studies. It is unclear why alveolar formation has a higher energy demand than saccule formation. Perhaps, construction of a more complex structure such as the alveolus requires additional energy consumption.

A limitation of our approach lies in the broad expression of Cre lines in multiple cell types over an extended developmental time. We propose that distinct cellular processes exhibit differential dependence on mitochondrial activity. While there is no evidence that a new epithelial cell type emerges from saccular to alveolar stages, the transcriptome of any cell type changes as development proceeds. The simplest model is that the same cell types with an altered transcriptome (dubbed cell subtypes) display different sensitivity to mitochondrial perturbation (and energy expenditure) as lung development proceeds from saccular to alveolar stages. This is based on the assumption that the subcellular events (e.g., ligand secretion and cellular contraction) that drive a cellular process (e.g., sacculation and secondary septation) are the main consumer of energy for a given cell subtype. However, it is also possible that the subcellular events that execute a given cellular process may not be the main consumer of energy. In this scenario, one would conclude that the phenotypic defects of a given cellular process due to compromised mitochondrial activity are indicative of the sensitivity of the subcellular events to mitochondrial activity. Finally, we cannot rule out the possibility that functions mediated by mitochondria other than energy production play an important role in a given cellular process (*Schumacker et al., 2014*). However, lack of apoptosis in the mutant lungs in our study suggests that mitochondria-controlled cell death does not contribute to the lung phenotypes.

Our work has focused on PDGF ligand secretion. Alveolar epithelial cells, especially AT1 cells, are the reservoirs for multiple ligands, including PDGFA, VEGFA, and SHH (*Zhang et al., 2020*; *Vila Ellis et al., 2020*; *Bellusci et al., 1997*). While other pathways have not been interrogated, different pathways may exhibit a varying degree of dependence on mitochondrial function. Similarly, this could provide a new way to functionally categorize signaling pathways on the basis of their energy demand in vivo. Such insight will provide a new blueprint of how cells dispense their energy source in tissues.

Secretion of the PDGF ligand from alveolar epithelial cells is impaired due to mitochondrial dysfunction in either activity or distribution. We surmise that the actomyosin cytoskeleton likely underlies this defect. However, it is also possible that the energy produced by mitochondria is required in many key steps of vesicular transport and membrane fusion. Additional insight would come from a careful assessment of the dependence of each process on mitochondrial function. Similarly, failure to power the actomyosin cytoskeleton in myofibroblasts due to disruption of mitochondrial activity or distribution

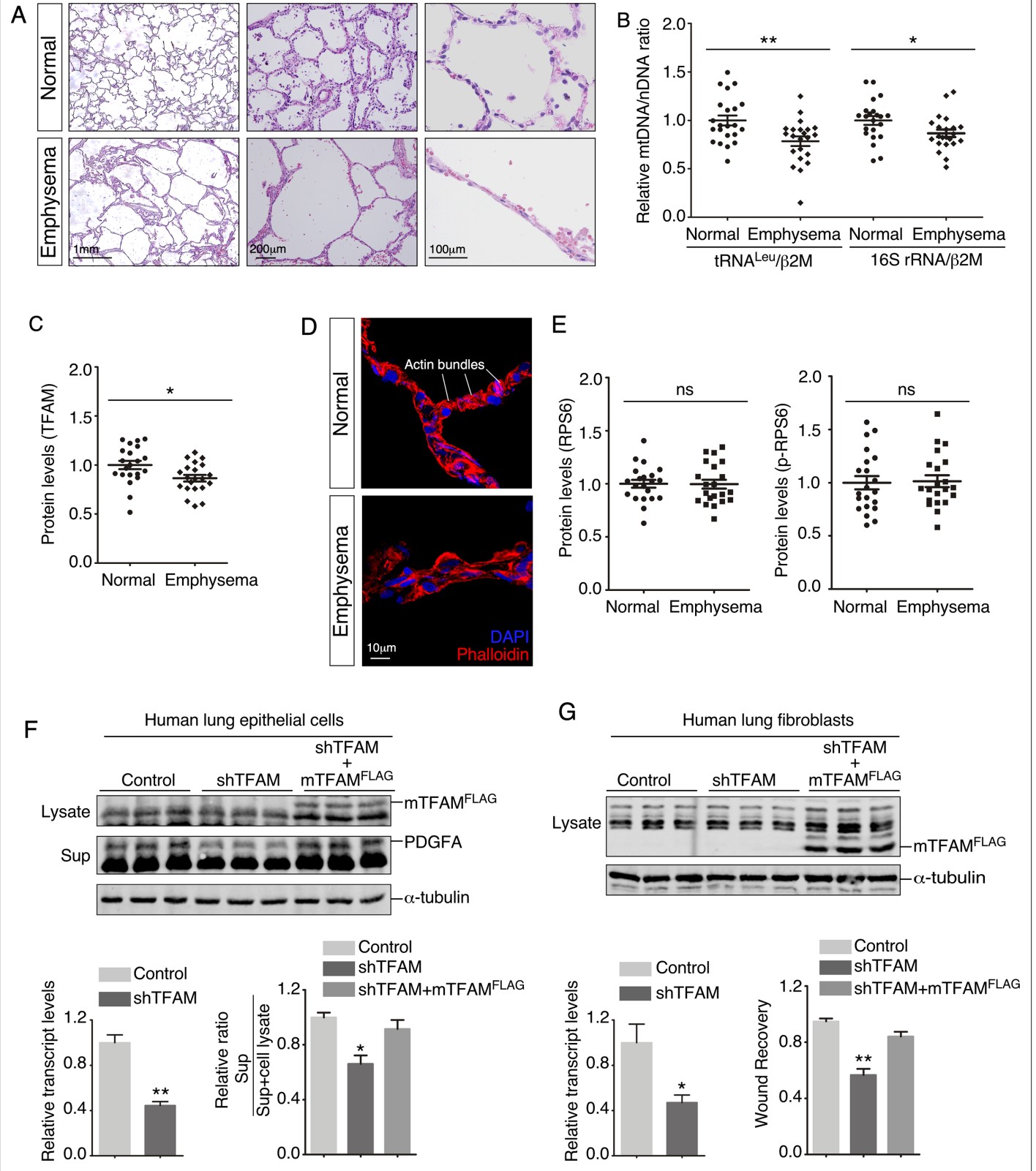

**Figure 8.** Lungs from emphysema patients exhibit a reduction in mitochondrial DNA and TFAM expression. (**A**) Hematoxylin and eosin-stained lung sections of normal and emphysema patients. Disruption of alveoli in emphysema patients resulted in enlarged airspace with thin primary septa. (**B**) qPCR analysis of mitochondrial DNA (mtDNA) to nuclear DNA (nDNA) in lung lysates derived from normal and emphysema patients (n = 22 for each group). mtDNA-encoded tRNA$^{Leu}$ and 16S rRNA and nDNA-encoded b2M (b2 microglobulin) were used in this study. The relative ratio of mtDNA to

*Figure 8 continued on next page*

*Figure 8 continued*

nDNA was calculated. (**C**) Quantification of TFAM protein levels in lung lysates derived from normal and emphysema patients (n = 21 for each group). (**D**) Immunostaining of lungs collected from control and emphysema patients. Phalloidin detected actin filaments. (**E**). Quantification of the protein levels of RPS6 and phosphorylated RPS6 (p-RPS6) in lung lysates derived from normal and emphysema patients (n = 21 for each group). (**F**) Western blot analysis of cell lysates and supernatants from human lung epithelial cells transduced with control constructs, constructs expressing shRNAs against human *TFAM* and constructs expressing mouse TFAM^FLAG (mTFAM^FLAG) as indicated. qPCR analysis revealed efficient knockdown of human *TFAM* transcripts. The amount of PDGFA released into the media was reduced in *TFAM*-knockdown cells compared to controls and was rescued by expression of mTFAM^FLAG (n = 3 for each group). α-Tubulin served as a loading control. (**G**) Western blot analysis of cell lysates from human lung fibroblasts transduced with control constructs, constructs expressing shRNAs against human *TFAM* and constructs expressing mouse TFAM^FLAG (mTFAM^FLAG) as indicated. qPCR analysis revealed efficient knockdown of human *TFAM* transcripts. Wound recovery assays using control, *TFAM*-knockdown fibroblasts, and *TFAM*-knockdown fibroblasts expressing mTFAM^FLAG were quantified. The migratory defect of fibroblasts was rescued by mTFAM^FLAG expression (n = 3 for each group). α-Tubulin served as a loading control. All values are mean ± SEM. *p<0.05; **p<0.01; ns, not significant (unpaired Student's *t*-test).

The online version of this article includes the following source data and figure supplement(s) for figure 8:

**Source data 1.** Relative mitochondrial DNA (mtDNA)/nuclear DNA (nDNA) ratio, protein levels, relative transcript levels, quantification of PDGFA secretion, and quantification of wound recovery.

**Figure supplement 1.** Mitochondria display inhomogeneous distribution in human lungs.

**Figure supplement 2.** TFAM protein levels are decreased in emphysema patients.

**Figure supplement 3.** RHOT1 controls PDGF secretion in human lung epithelial cells and migration of human lung fibroblasts.

**Figure supplement 3—source data 1.** Relative transcript levels, quantification of PDGFA secretion, and quantification of wound recovery.

could cause the contraction and/or migratory defect. ATP produced by mitochondria is known for the assembly of the actomyosin cytoskeleton. By contrast, how regulation of mitochondrial distribution is superimposed upon the formation of the actomyosin cytoskeleton is less understood at the molecular level. This scenario is further complicated by the observation that F-actin and intermediate filaments also play a key role in mitochondrial dynamics and functions (*Shah et al., 2021*). Technical advances that enable live imaging (*Looney and Bhattacharya, 2014*) of mitochondria (*Gökerküçük et al., 2020*) and the cytoskeleton would provide new tools to address this important issue.

We have uncovered the role of mitochondria in alveolar epithelial cells and fibroblasts/myofibroblasts during alveolar formation. Whether the function of endothelial cells/pericytes or other cell types not yet tested also relies on mitochondrial activity and/or distribution in this process is unknown (*Ding et al., 2011*; *Swonger et al., 2016*). This would require future studies using an approach similar to that employed in this work. Again, the dependence of cell types, cellular processes, and signaling pathways on mitochondrial function can only be revealed through studies in vivo.

We envision that a complex regulatory network must be in place to regulate mitochondrial number and distribution in distinct cellular processes. Our work shows that mTOR complex 1 is a key player in controlling mitochondrial function during alveolar formation. This is consistent with the role of mTORC1 in regulating mitochondrial function in cell-based assays. Identifying additional components in the signaling network would reveal the key hubs in the signaling network that control mitochondrial function during alveologenesis.

Mitochondrial copy number and TFAM expression levels are reduced in the lungs of COPD/emphysema patients. Moreover, knockdown of *TFAM* or *RHOT1* in human lung epithelial cells or fibroblasts impairs their cellular properties, including PDGF secretion and myofibroblast migration. We speculate that changes in cellular properties due to disturbed mitochondrial activity and distribution contribute to the pathogenesis of COPD/emphysema (*Ryter et al., 2018*; *Caldeira et al., 2021*; *Hara et al., 2018*; *Cloonan and Choi, 2016*). Disturbance of mitochondrial function and cellular properties could be related to inflammatory responses in COPD/emphysema. To further test this idea would rely on the analysis of lungs at different stages of disease progression with a focus on identifying cellular changes that are directly related to mitochondrial dysfunction. It also necessitates new approaches that enable a mechanistic correlation between mitochondrial function and disease pathogenesis and progression. For instance, analysis of the transcriptome and proteome of various lung cell types at single-cell levels could reveal alterations in a subset of cells that herald the process of emphysematous changes. Such analysis could also uncover changes in the signaling network that connects mitochondria to cellular processes.

Taken together, our work has yielded new molecular insight into the old question of energy utilization in vivo. In particular, energy production by mitochondria is channeled in a spatially specific

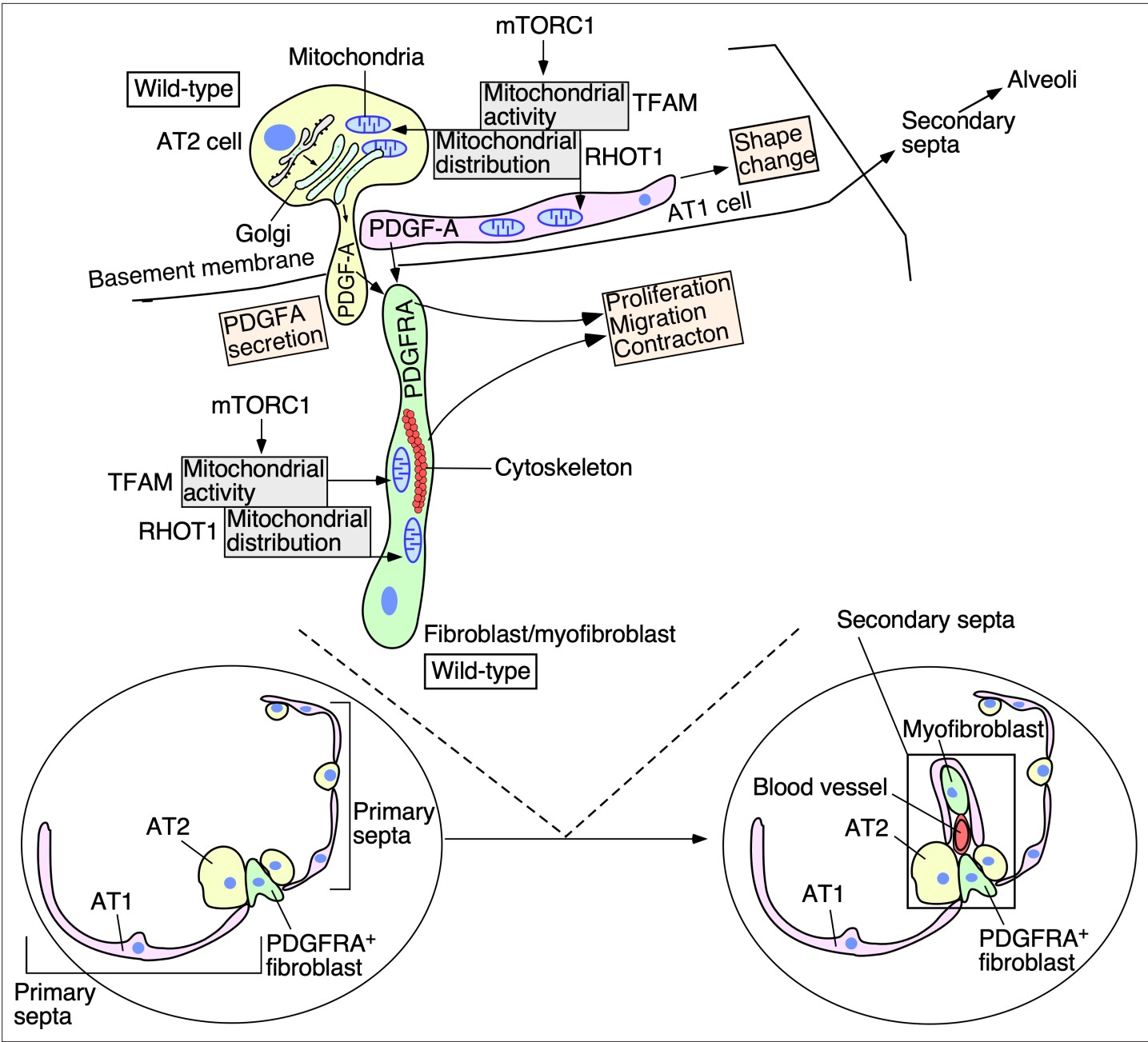

**Figure 9.** A model of regulating alveolar formation through mitochondrial activity and distribution. The main players during alveolar formation are shown in the schematic diagram. Mitochondria display dynamic subcellular distribution in alveolar epithelial cells and mesenchymal fibroblasts/myofibroblasts. *Tfam* controls mitochondrial activity while *Rhot1* regulates mitochondrial distribution. Mitochondrial activity and distribution in alveolar epithelial cells (type I [AT1] and type II [AT2]) contribute to secretion of the PDGF-A ligand. Reception of PDGF-A by mesenchymal fibroblasts/myofibroblasts is critical to fibroblast/myofibroblast proliferation and migration, a key step in secondary septation. Similarly, mitochondrial activity and distribution in fibroblasts/myofibroblasts are also required for fibroblast/myofibroblast contraction and migration, likely through powering the cytoskeleton. The mTORC1 pathway plays a central role in controlling mitochondrial function during alveolar formation. Specification of alveolar epithelial cells and fibroblasts/myofibroblasts was unaffected in mutant mouse lungs in which mitochondrial activity and distribution were perturbed. We propose that essential cellular processes have differential requirements of mitochondrial activity and distribution. Investigating how mitochondria control signaling pathways and cellular processes in vivo provide a new way to functionally define signaling pathways and cellular processes. We surmise that the regulatory circuitry mediated by mitochondria activity and distribution is also deployed during alveolar repair following lung injury and could contribute to the pathogenesis of chronic obstructive lung disease (COPD)/emphysema.

manner to power cellular machinery and drive cellular processes in distinct cell types. Diverse cell types and cellular processes have a unique energy demand, which is likely executed by a signaling network. These investigations form the basis of additional studies to explore this new concept in alveolar formation and repair and in other physiological and pathological processes in vivo.

## Materials and methods

### Animal husbandry

Mouse strains used in this study are listed in the Key resources table. All mouse experiments described in this study were performed according to the protocol approved by the Institutional Animal Care and Use Committee (IACUC) of the University of California, San Francisco (UCSF).

### Tamoxifen administration

Tamoxifen was prepared by dissolving in corn oil to a concentration of 50 mg/ml (*Zhang et al., 2020*). For postnatal (P) injection, the tamoxifen stock was diluted 1/10 in corn oil to make a final concentration of 5 mg/ml. 50 µl of tamoxifen was delivered through oral gavage or direct injection into the stomach of neonatal mice.

### Measurement of MLI

Measurement of MLI was performed as previously reported (*Zhang et al., 2020*). Briefly, 15 fields without visible blood vessels or airways from three different histological sections per animal were captured at ×20 magnification using a Nikon Eclipse E1000 microscope. A grid with horizontal and vertical lines (10 each) was superimposed on the images using ImageJ. The MLI (Lm) was calculated as Lm = L/N, where L is the total length of horizontal plus vertical lines, and N is the total number of the intercepts.

### Measurement of alveolar wall thickness

To measure the thickness of the primary septal wall (*Osterreicher et al., 1999*), 15 representative fields from histological sections of lungs at the indicated age were imaged at ×100 magnification using a Nikon Eclipse E1000 microscope. Three vertical lines were drawn on each image such that they can cross the primary septal wall. The thickness of the primary septal wall was evaluated by ImageJ as the vertical distance of the line that traversed the primary septal wall.

### Whole-lung imaging, histology, and immunohistochemistry

To image the whole lungs that carried GFP or RFP, dissected mouse lungs at the indicated stages were placed under a Nikon Eclipse E1000 microscope equipped with a SPOT 2.3 CCD camera.

Mouse lungs at indicated time points were collected and fixed with 4% paraformaldehyde (PFA) in PBS on ice for 1 hr. The tissues were embedded in paraffin wax or OCT and sectioned at 7 µm. For histological analysis of lung sections, hematoxylin and eosin (H&E) staining was performed as previously described (*Zhang et al., 2020*; *Lin et al., 2017*). Images were taken using a SPOT 2.3 CCD camera connected to a Nikon Eclipse E1000 microscope.

To detect PDGFA in lung cells, lungs from $Sox9^{Cre/+}$; $Pdgfa^{ex4COIN/+}$ (control) and $Tfam^{f/f}$; $Sox9^{Cre/+}$; $Pdgfa^{ex4COIN/+}$ mice at P3 were dissected and fixed in 4% PFA on ice for 1 hr. Lungs were washed in 0.02% NP40 in PBS for 2 hr, then placed in X-gal staining solution (5 mM $K_3Fe(CN)_6$, 5 mM $K_4Fe(CN)_6$, 2 mM $MgCl_2$, 0.01% sodium deoxycholate, 0.02% NP-40, 1 mg/ml X-gal) for 72 hr at 37°C. The stained lungs were paraffin embedded and sectioned. Images were taken using a SPOT 2.3 CCD camera connected to a Nikon Eclipse E1000 microscope.

Immunofluorescence was performed as previously described (*Lin et al., 2017*). Antibodies used in this study are listed in the Key resources table. The primary antibodies used for wax sections were: chicken anti-GFP (1:200, Abcam, Cat# ab13970), rabbit anti-NKX2.1 (1:100, Epitomics, Cat# 2044-1), goat anti-CC10 (1:200, Santa Cruz Biotechnology, Cat# sc-9773), mouse anti-acetylated tubulin (1:200, Sigma-Aldrich, Cat# T6793), rabbit anti-prosurfactant protein C (proSP-C) (1:200, MilliporeSigma, Cat# AB3786), hamster anti-T1α (1:200, Developmental Studies Hybridoma Bank, Cat# 8.1.1), and mouse anti-HOPX (1:100, Santa Cruz Biotechnology, Cat# sc-398703). The primary antibodies used for frozen sections were rabbit anti-MPC1 (1:100, Millipore/Sigma, Cat#

HPA045119), rat anti-E-cadherin (1:200, Invitrogen, Cat# 13-1900), mouse anti-MTCO1 (1:100, Abcam, Cat# ab14705), chicken anti-GFP (1:300, Abcam, Cat# ab13970), mouse anti-ACTA2 (1:200, Thermo Scientific Lab Vision, Cat# MS-113-P0), rat anti-PECAM-1 (CD31) (1:150, Santa Cruz Biotechnology, Cat# sc-18916), rabbit anti-PDGFRA (1:150, Cell Signaling Technology, Cat# 3164), rabbit anti-phospho-PDGFRA (Tyr754) (1:100, Cell Signaling Technology, Cat# 2992), mouse anti-S6 ribosomal protein (1:100, Cell Signaling Technology, Cat# 2317), and rabbit anti-phospho-S6 ribosomal protein (Ser235/236) (1:100, Cell Signaling Technology, Cat# 4856). Secondary antibodies and conjugates used were donkey anti-rabbit Alexa Fluor 488 or 594 (1:1000, Life Technologies), donkey anti-chicken Alexa Fluor 488 or 647 (1:1000, Life Technologies), donkey anti-mouse Alexa Fluor 488 or 594 (1:1000, Life Technologies), and donkey anti-rat Alexa Fluor 594 (1:1000, Life Technologies). The biotinylated secondary antibodies used were goat anti-hamster (1:1000, Jackson ImmunoResearch Laboratories), donkey anti-rabbit (1:1000, Jackson ImmunoResearch Laboratories), donkey anti-rat (1:1000, Jackson ImmunoResearch Laboratories), and horse anti-mouse (1:1000, Jackson ImmunoResearch Laboratories). The signal was detected using streptavidin-conjugated Alexa Fluor 488, 594, or 647 (1:1000, Life Technologies) or HRP-conjugated streptavidin (1:1000, PerkinElmer) coupled with fluorogenic substrate Alexa Fluor 594 tyramide for 30 s (1:200, TSA kit; PerkinElmer). F-actin was stained with rhodamine-conjugated phalloidin (1:200, Sigma) in PBS for 2 hr. Since the filamentous actin is sensitive to methanol, ethanol, and high temperature, we only used OCT-embedded frozen sections for F-actin staining.

Confocal images were captured on a Leica SPE laser-scanning confocal microscope. Adjustment of red/green/blue/gray histograms and channel merges were performed using LAS AF Lite (Leica Microsystems).

## Fibroblast/myofibroblast proliferation assays

The rate of cell proliferation was determined through EdU incorporation as previously described (*Zhang et al., 2020*). Mouse pups at the indicated time points were intraperitoneally injected with EdU/PBS solution for 1 hr before dissection. The Click-iT EdU Alexa Fluor 488 Imaging Kit (Life Technologies) was used to quantify EdU incorporation. The sections were co-stained with antibody against PDGFRA. Cell proliferation rate was calculated as the ratio of (EdU+ PDGFRA+ cells)/(PDGFRA+ cells).

## Fibroblast/myofibroblast migration assay in vitro

The migratory capacity of lung fibroblasts/myofibroblasts was assessed by the Culture-Insert 2 Well system (ibidi) (*Zhang et al., 2020*). Briefly, at P3, dissected lungs from control, *Tfam*^f/f; *Pdgfra*^Cre/+ and *Rhot1*^f/f; *Pdgfra*^Cre/+ mice were minced into small pieces and digested in solution (1.2 U/ml dispase, 0.5 mg/ml collagenase B, and 50 U/ml DNase I), rocking at 37°C for 2 hr to release single cells. After adding an equal volume of culture medium (DMEM with 10% fetal bovine serum (FBS), 2× penicillin/streptomycin, and 1× L-glutamine), the samples were filtered through 40 μm cell strainers and centrifuged at 600 × *g* for 10 min. The dissociated cells were resuspended in 200 μl of culture medium and plated into wells (100 μl per well). The lung fibroblasts/myofibroblasts were allowed to attach to the fibronectin-coated plates for 2–3 hr. Fresh culture medium was added, and the attached fibroblasts/myofibroblasts were cultured 2 or 3 days to reach 100% confluence. The confluent fibroblasts/myofibroblasts were switched to starvation medium (DMEM with 0.5% FBS and 1× penicillin/streptomycin) for 16 hr before removal of the insert. Fibroblasts/myofibroblasts migration was assessed 36–48 hr afterward.

## Lentiviral production and transduction

Lentiviruses were produced in HEK293T cells maintained in DMEM containing 10% FBS, 1× penicillin/streptomycin, and 1× L-glutamine (*Zhang et al., 2020*). HEK293T cells were plated and transfected when they reached 70% confluence on the following day. 2 μg of pMD2.G, 2 μg of psPAX2, 4 μg of PDGFA-3xFLAG (in the modified pSECC lentiviral vector), and 50 μl of polyethylenimine (PEI) (1 μg/μl) were mixed in 1000 μl OPTI-MEM and added to a 10 cm dish. Incubation medium was replaced 1 day after transfection. 48 hr post-transfection, the viral supernatant was collected, filtered through 0.45 μm PVDF filter, then added to primary lung cells together with 8 μg/ml polybrene. 12 hr post-transduction, the medium was replaced with fresh culture medium.

## shRNA-mediated gene silencing

shRNAs against the 3'UTR of human *TFAM* and *RHOT1* were designed by pSicOligomaker (*Reynolds et al., 2004*). Paired oligonucleotides were annealed and inserted into the pLentiLox3.7 lentiviral vector. The packaging vectors used were either the pLP1/pLP2/pLP/VSV-G or pMD2.G/psPAX2 combinations. Primers used for shRNAs were shTFAM-1 forward,

5'-TGGTGCTGAGGAGTGTTAAATTCAAGAGATTTAACACTCCTCAGCACCTTTTTTC-3'; reverse, 5' TCGAGAAAAAAGGTGCTGAGGAGTGTTAAATCTCTTGAATTTAACACTCCTCAGCACCA-3', shTFAM-2 forward, 5'-TGACTTCTGCCAGCATAATATTCAAGAGATATTATGCTGGCAGAAGTCTTTTT TC-3'; reverse, 5'-TCGAGAAAAAAGACTTCTGCCAGCATAATATCTCTTGAATATTATGCTGGCAG AAGTCA-3', shTFAM-3 forward, 5'-TGTACTCTTGTTTCCTTATATTCAAGAGATATAAGGAAACAAGAG TACTTTTTTC-3'; reverse, 5'-TCGAGAAAAAAGTACTCTTGTTTCCTTATATCTCTTGAATATAAGGAAA CAAGAGTACA-3', shRHOT1-1 forward, 5'-TGAAACAGCGATGATATAAATTCAAGAGATTTATATC ATCGCTGTTTCTTTTTTC-3'; reverse, 5'-TCGAGAAAAAAGAAACAGCGATGATATAAATCTCTTGAATT TATATCATCGCTGTTTCA-3', shRHOT1-2 forward, 5'-TGTACATTCTGAATGCTTTATTCAAGAGATAAA GCATTCAGAATGTACTTTTTTC-3'; reverse, 5'-TCGAGAAAAAAGTACATTCTGAATGCTTTATCTCTTG AATAAAGCATTCAGAATGTACA-3', shRHOT1-3 forward, 5'-TGAAATGATGTTTCTAGACATTCAA GAGATGTCTAGAAACATCATTTCTTTTTTC-3'; reverse, 5'-TCGAGAAAAAAGAAATGATGTTTCT AGACATCTCTTGAATGTCTAGAAACATCATTTCA-3'.

## PDGFA secretion assay

PDGFA secretion assay was performed as previously described (*Zhang et al., 2020*). In brief, PDGFA-3xFLAG was stably expressed in control and *Tfam*- and *Rhot1*-deficient primary lung cells (derived from the lungs of control, *Tfam^f/f^; Pdgfra^Cre/+^* and *Rhot1^f/f^; Pdgfra^Cre/+^* mice, respectively) that were plated onto 10 cm dishes. The culture media were replaced with OPTI-MEM supplemented with insulin, transferrin, and selenium (ITS) once the cells reached 100% confluence. 24 hr post-incubation, the supernatants were mixed with 1× protein inhibitor cocktail, filtered through 0.45 µm filters, and centrifuged at high speed (>12,000 rpm) for 15 min at 4°C. The filtrates were then concentrated in protein concentration columns (Millipore CENTRICON YM-10 Centrifugal Filter Unit 2 ml 10 kDa) through centrifugation at 2000 × *g* for 1 hr at 4°C. Concentrated supernatants were mixed with immunoprecipitation (IP) buffer (50 mM Tris pH 7.4, 2 mM EDTA, 150 mM NaCl, 0.5% Triton X-100, 1× protein inhibitor cocktail) in a total volume of 500 µl. Meanwhile, cells seeded on the plate were harvested and lysed in IP buffer. Immunoprecipitation was performed using FLAG-M2 beads following the standard procedure. Western blotting was performed to detect PDGFA in the supernatants and lysates derived from control, *Tfam*-deficient, and *Rhot1*-deficient primary lung cells.

## qPCR analysis

qPCR was performed as previously described (*Zhang et al., 2020*). Briefly, the right cranial lobe was dissected from the mouse lungs of the indicated genotypes and time points, and homogenized in 1 ml TRIzol (Life Technologies). The homogenates were added to 200 µl chloroform, and then centrifuged for 15 min at 12,000 rpm. The upper aqueous layer was collected and mixed with an equal volume of 70% ethanol. RNAs were extracted with the RNeasy Mini Kit (QIAGEN) following the manufacturer's instructions. The extracted RNAs were reverse-transcribed with the Maxima First Strand cDNA Synthesis Kit (Thermo Scientific). Quantitative PCR (qPCR) was performed on the Applied Biosystems QuantStudio 5 Real-Time PCR System. Primers used for qPCR were mouse *Pdgfa* forward,

5'-CTGGCTCGAAGTCAGATCCACA-3'; reverse, 5'-GACTTGTCTCCAAGGCATCCTC-3', mouse *Pdgfra* forward, 5'-TGCAGTTGCCTTACGACTCCAGAT-3'; reverse, 5'-AGCCACCTTCATTACAGGTT GGGA-3', mouse *Acta2* forward, 5'-ATGCAGAAGGAGATCACAGC-3'; reverse, 5'-GAAGGTAGACAG CGAAGCC-3', mouse *Eln* forward, 5'-GCCAAAGCTGCCAAATACG-3'; reverse, 5'-CTCCAGCTCCAA CACCATAG-3', mouse *Gapdh* forward, 5'-AGGTTGTCTCCTGCGACTTCA-3'; reverse, 5'-CCAGG AAATGAGACAAAGTT-3'.

## Analysis of mtDNA/nDNA ratio

Lung samples from mice and human patients were lysed in lysis buffer (100 mM Tris pH 7.5, 5 mM EDTA, 0.4% SDS, 200 mM NaCl, 50 µg/ml proteinase K), and incubated in a 55°C chamber overnight. The tissues were digested with 100 µg/ml RNase A at 37°C for 30 min to degrade the RNAs. An equal

volume of phenol/chloroform was added to the lysed samples, which were centrifuged at 12,000 rpm for 10 min. Lung DNAs were concentrated by ethanol precipitation. For mouse lungs, the mitochondrial 16S rRNA or ND1 gene and the nuclear *Hk2* gene were used to calculate the relative ratio of mitochondrial (mt) to nuclear (n) DNA copy number. For human lungs, the mitochondrial tRNA$^{Leu(UUR)}$ or 16S rRNA gene and the nuclear β2-microglobulin (*β2M*) gene were employed to determine the relative ratio of mtDNA to nDNA. qPCR was performed on the Applied Biosystems QuantStudio 5 Real-Time PCR System. The primer pairs used for the indicated genes were: mouse 16S rRNA forward, 5'-CCGCAAGGGAAAGATGAAAGAC-3'; reverse, 5'-TCGTTTGGTTTCGGGGTTTC-3', mouse mt-ND1 forward, 5'-CTAGCAGAAACAAACCGGGC-3'; reverse, 5'-CCGGCTGCGTATTCTACGTT-3', mouse Hk2 forward, 5'-GCCAGCCTCTCCTGATTTTAGTGT-3'; reverse, 5'-GGGAACACAAAAGACCTCTTC TGG-3', human tRNA$^{Leu(UUR)}$ forward, 5'-CACCCAAGAACAGGGTTTGT-3'; reverse, 5'-TGGCCATGG GTATGTTGTTA-3', human 16S rRNA forward, 5'-GCCTTCCCCCGTAAATGATA-3'; reverse, 5'-TTATG CGATTACCGGGCTCT-3', human β-2-microglobulin (*β2M*) forward, 5'-TGCTGTCTCCATGTTTGATGT ATCT-3'; reverse, 5'-TCTCTGCTCCCCACCTCTAAGT-3'.

## Enzymatic activities of the mitochondrial electron transport chain (ETC) complex

Measurement of the enzymatic activity of mitochondrial ETC complex was performed as previously described (*Masand et al., 2018*). In brief, frozen lung tissues from control, *Tfam^{f/f}; Pdgfra^{Cre/+}*, *Tfam^{f/f}; Sox9^{Cre/+}*, *Rhot1^{f/f}; Pdgfra^{Cre/+}* and *Rhot1^{f/f}; Sox9^{Cre/+}* mice at P5 were homogenized in homogenization buffer (120 mM KCl, 20 mM HEPES, 1 mM EGTA, pH 7.4) using a Dounce homogenizer. The Pierce BCA Protein Assay Kit was used to determine protein concentrations before spectrophotometric kinetic assays. Complex I (NADH:ubiquinone oxidoreductase) activity was determined by measuring the oxidation of NADH at 340 nm in a reaction mixture (50 mM potassium phosphate [pH 7.5], 1.7 mM potassium ferricyanide, 0.2 mM NADH, and homogenate of interest). Potassium ferricyanide was used as the electron acceptor for Complex I. Complex IV (cytochrome *c* oxidase) activity was determined by measuring the oxidation of cytochrome *c* at 550 nm in a reaction solution (50 mM potassium phosphate [pH 7.0], 100 μM reduced cytochrome *c,* and homogenate of interest).

## Mitochondrial network

Fibroblasts were derived from control, *Tfam^{f/f}; Pdgfra^{Cre/+}* and *Rhot1^{f/f}; Pdgfra^{Cre/+}* lungs and transduced with lentiviral constructs expressing Mito-7-mEmerald to label mitochondria. After 48 hr, cells were fixed and co-stained with MTCO1. Tubular mitochondria are the dominant form in control fibroblasts, while short tubular and fragmented mitochondria could be found in *Tfam*-deficient fibroblasts. Fragmented mitochondria are characterized as dot-like structures; the morphology of short tubular mitochondria is intermediate between tubular and fragmented mitochondria.

## Human lung tissues

Human lung samples were processed as previously described (*Zhang et al., 2020*). Briefly, lung tissues were obtained from severe emphysema (Global Initiative for Chronic Obstructive Lung Disease Criteria, stages III or IV) at the time of lung transplantation. The donor control lung samples were indicated physiologically and pathologically normal (*Ware et al., 2002*). Written informed consent was obtained from all subjects, and the study was approved by the University of California, San Francisco Institutional Review Board (IRB approval # 13-10738).

## Human lung cells

The human lung epithelial cell line (1310 cell line) was derived from human alveolar type II cells. Cells were immortalized by lentiviral transduction with hTERT and CDK4 to create a stable cell line (*Ramirez et al., 2004*).

Human airway fibroblasts were derived from donor lungs not utilized for lung transplantation. It was approved by the Committee on Human Research at the University of California, San Francisco, in full accordance with the Declaration of Helsinki principles. Airway fibroblasts were cultured from the lung parenchyma by the explant technique and used passages 1–4, as previously described (*Araya et al., 2007*). The identity of the cell lines has been confirmed by STR profiling, and no mycoplasma contamination was found.

## Western blotting analysis

Human lung tissues from normal subjects and emphysema patients and mouse lung samples were homogenized in RIPA buffer with 1× Protease Inhibitor Cocktail and 1× PMSF. The lysates were centrifuged at full speed for 15 min at 4°C and analyzed by Western blotting as previously described (*Lin et al., 2017*). The primary antibodies used were rabbit anti-TFAM (1: 2000, Proteintech, Cat# 22586-1-AP), mouse anti-S6 ribosomal protein (1:2000, Cell Signaling Technology, Cat# 2317), rabbit anti-phospho-S6 ribosomal protein (Ser235/236) (1:2000, Cell Signaling Technology, Cat# 4856), mouse anti-OPA1 (clone 18) (1:2000, BD Transduction Laboratories, Cat# 612606), mouse anti-DLP1 (1:2000, BD Transduction Laboratories, Cat# 611113), rabbit anti-phospho-DRP1 (Ser616) (1:1000, Cell Signaling Technology, Cat# 3455S), and mouse anti-alpha-tubulin (1:3000, Developmental Studies Hybridoma Bank, Cat# 12G10).

## Statistical analysis

All the statistical comparisons between different groups are shown as mean value ± SEM. The p-values were calculated by two-tailed Student's *t*-tests, and statistical significance was evaluated as $*p<0.05$; $**p<0.01$; $***p<<0.001$. More than or equal to three biological repeats were performed, and the detailed biological replicates (n numbers) are indicated in the figure legends.

## Acknowledgements

Some data for this study were acquired at the Nikon Imaging Center at CVRI. This work was supported by R01 HL142876 from the National Institutes of Health to PTC.

## Additional information

### Funding

| Funder | Grant reference number | Author |
| --- | --- | --- |
| National Heart, Lung, and Blood Institute | R01 HL142876 | Pao-Tien Chuang |

The funders had no role in study design, data collection and interpretation, or the decision to submit the work for publication.

### Author contributions

Kuan Zhang, Pao-Tien Chuang, Conceptualization, Data curation, Formal analysis, Funding acquisition, Investigation, Project administration, Supervision, Validation, Visualization, Writing - original draft, Writing - review and editing; Erica Yao, Conceptualization, Data curation, Formal analysis, Investigation, Validation, Visualization, Writing - original draft, Writing - review and editing; Biao Chen, Ethan Chuang, Julia Wong, Paul J Wolters, Data curation, Formal analysis, Validation, Visualization, Writing - review and editing; Robert I Seed, Stephen L Nishimura, Data curation, Validation, Visualization

### Author ORCIDs

Kuan Zhang http://orcid.org/0000-0002-9723-1328
Pao-Tien Chuang http://orcid.org/0000-0002-8961-8653

### Ethics

This study was performed in strict accordance with the recommendations in the Guide for the Care and Use of Laboratory Animals of the National Institutes of Health. All of the animals were handled according to approved institutional animal care and use committee (IACUC) protocols of the University of California, San Francisco. The protocol was approved by the Committee on the Ethics of Animal Experiments of the University of California, San Francisco (Approval Number: AN187712-01).

### Decision letter and Author response

Decision letter https://doi.org/10.7554/eLife.68598.sa1
Author response https://doi.org/10.7554/eLife.68598.sa2

## Additional files

### Supplementary files
• Transparent reporting form

### Data availability
All data generated or analysed during this study are included in the manuscript and supporting files. This study does not generate source data files that need to be deposited.

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

# Appendix 1

## Appendix 1—key resources table

| Reagent type (species) or resource | Designation | Source or reference | Identifiers | Additional information |
|---|---|---|---|---|
| Antibody | Anti-ACTA2 (mouse monoclonal) | Thermo Scientific Lab Vision | Cat# MS-113-P0; RRID:AB_64001 | IF (1:200) |
| Antibody | Anti-CC10 (goat polyclonal) | Santa Cruz Biotechnology | Cat# sc-9773; RRID:AB_2183391 | IF (1:200) |
| Antibody | Anti-DLP1 (mouse monoclonal) | BD Biosciences | Cat# 611113; RRID:AB_398424 | WB (1:2000) |
| Antibody | Anti-phospho-DRP1 (Ser616) (rabbit polyclonal) | Cell Signaling Technology | Cat# 3455; RRID:AB_2085352 | WB (1:1000) |
| Antibody | Anti-E-cadherin (rat monoclonal) | Invitrogen | Cat# 13-1900; RRID:AB_2533005 | IF (1:200) |
| Antibody | Anti-GFP (chicken polyclonal) | Abcam | Cat# ab13970; RRID:AB_300798 | IF (1:200) |
| Antibody | Anti-mouse HOPX (mouse monoclonal) | Santa Cruz Biotechnology | Cat# sc-398703; RRID:AB_2687966 | IF (1:100) |
| Antibody | Anti-MPC1 (BRP44L) (rabbit polyclonal) | MilliporeSigma | Cat# HPA045119; RRID:AB_10960421 | IF (1:100) |
| Antibody | Anti-MTCO1 [1D6] (mouse monoclonal) | Abcam | Cat# ab14705; RRID:AB_2084810 | IF (1:100) |
| Antibody | Anti-NKX2.1 (rabbit monoclonal) | Epitomics | Cat# 2044-1; RRID:AB_1267367 | IF (1:100) |
| Antibody | Anti-OPA1, clone 18 (mouse monoclonal) | BD Biosciences | Cat# 612606; RRID:AB_399888 | WB (1:2000) |
| Antibody | Anti-PDGF receptor alpha (rabbit polyclonal) | Cell Signaling Technology | Cat# 3164; RRID:AB_2162351 | IF (1:150) |
| Antibody | Anti-phospho-PDGF receptor $\alpha$ (Tyr754) (23B2) (rabbit monoclonal) | Cell Signaling Technology | Cat# 2992; RRID:AB_390728 | IF (1:100) |
| Antibody | Anti-PECAM-1 (MEC 13.3) (rat monoclonal) | Santa Cruz Biotechnology | Cat# sc-18916; RRID:AB_627028 | IF (1:150) |
| Antibody | Anti-prosurfactant protein C (proSP-C) (rabbit polyclonal) | MilliporeSigma | Cat# AB3786; RRID:AB_91588 | IF (1:200) |
| Antibody | Anti-S6 ribosomal protein (54D2) (mouse monoclonal) | Cell Signaling Technology | Cat# 2317; RRID:AB_2238583 | IF (1:100); WB (1:2000) |
| Antibody | Anti-phospho-S6 ribosomal protein (Ser235/236)(2F9) (rabbit monoclonal) | Cell Signaling Technology | Cat# 4856; RRID:AB_2181037 | IF (1:100); WB (1:2000) |
| Antibody | Anti-T1$\alpha$ (hamster monoclonal) | Developmental Studies Hybridoma Bank | Cat# 8.1.1; RRID:AB_531893 | IF (1:200) |
| Antibody | Anti-TFAM (rabbit polyclonal) | Proteintech | Cat# 22586-1-AP; RRID:AB_11182588 | WB (1:2000) |
| Antibody | Anti-Rat TGN38 (sheep polyclonal) | Bio-Rad Laboratories | Cat# AHP499G; RRID:AB_2203272 | IF (1:100) |
| Antibody | Anti-acetylated $\alpha$-tubulin, clone 6-11B-1 (mouse monoclonal) | MilliporeSigma | Cat# T6793; RRID:AB_477585 | IF (1:200) |
| Antibody | Anti-alpha-tubulin (mouse monoclonal) | Developmental Studies Hybridoma Bank | Cat# 12G10; RRID:AB_1157911 | WB (1:3000) |
| Chemical compound, drug | ANTI-FLAG M2 Affinity Gel | Sigma-Aldrich | Cat# A2220; RRID:AB_10063035 | |
| Chemical compound, drug | Biotin-XX Phalloidin | Molecular Probes | Cat# B7474 | |
| Chemical compound, drug | DMEM | Mediatech | Cat# 10-013-CV | |
| Chemical compound, drug | Fetal bovine serum (FBS) | Gibco | Cat# 10437-028 | |
| Chemical compound, drug | Fibronectin | Corning | Cat# 354008 | |

*Appendix 1 Continued on next page*

*Appendix 1 Continued*

| Reagent type (species) or resource | Designation | Source or reference | Identifiers | Additional information |
|---|---|---|---|---|
| Chemical compound, drug | Glutaraldehyde, 8% aqueous solution, EM grade | Electron Microscopy Sciences | Cat# 16000 | |
| Chemical compound, drug | Insulin, transferrin, and selenium (ITS) | Gibco | Cat# 51300-044 | |
| Chemical compound, drug | Paraformaldehyde, 16% solution, EM grade | Electron Microscopy Sciences | Cat# 15700 | |
| Chemical compound, drug | Paraformaldehyde | Sigma-Aldrich | Cat# P6148 | |
| Chemical compound, drug | Polyethylenimine (PEI) | Polysciences, Inc | Cat# 23966-2 | |
| Chemical compound, drug | Penicillin/streptomycin | Gibco | Cat# 15070-063 | |
| Chemical compound, drug | Protease Inhibitor Cocktail | Biotool | Cat# B14001 | |
| Chemical compound, drug | Rhodamine-conjugated phalloidin | Molecular Probes | Cat# R415 | |
| Chemical compound, drug | Tamoxifen | Toronto Research Chemicals | Cat# T006000 | |
| Chemical compound, drug | TRIzol Reagent | Ambion | Cat# 15596018 | |
| Cell line (*Homo sapiens*) | Human lung epithelial cells (1310 cell line) | John D. Minna (***Ramirez et al., 2004***) | | |
| Cell line (*H. sapiens*) | Human lung airway fibroblasts | Stephen L. Nishimura (***Araya et al., 2007***) | | |
| Commercial assay or kit | Click-iT EdUAlexa Fluor 488 Imaging Kit | Thermo Fisher | Cat# C10337 | |
| Commercial assay or kit | Centriprep Ultracel YM-10 Centrifugal Filter Devices | MilliporeSigma | Cat# 4305 | |
| Commercial assay or kit | Maxima First Strand cDNA Synthesis Kit | Thermo Scientific | Cat# K1641 | |
| Commercial assay or kit | RNeasy Mini Kit | QIAGEN | Cat# 74104 | |
| Commercial assay or kit | TSA Plus Cyanine 3 (Cy3) Fluorescein detection kit | PerkinElmer | Cat# NEL753001KT | |
| Commercial assay or kit | Two-well culture insert | Ibidi | Cat# 80209 | |
| Genetic reagent (*Mus musculus*) | $CAGG^{Cre}\text{-}ER^{TM}$ [B6.Cg-Tg(CAG-cre/Esr1*)5Amc/J] | The Jackson Laboratory | Stock# 004682; RRID:IMSR_JAX:004682 | |
| Genetic reagent (*M. musculus*) | $Cd4^{Cre}$ [B6.Tg(Cd4-cre)1Cwi] | The Jackson Laboratory | Stock# 022071; RRID:MGI:3691126 | |
| Genetic reagent (*M. musculus*) | $Cx3cr1^{CreER}$ [B6.129P2(C)-$Cx3cr1^{tm2.1(cre/ERT2)Jung}$/J] | The Jackson Laboratory | Stock# 020940; RRID:IMSR_JAX:020940 | |
| Genetic reagent (*M. musculus*) | $Dermo1^{Cre}$ [$Twist2^{tm1.1(cre)Dor}$/J] | David Ornitz (***Yu et al., 2003***) | | |
| Genetic reagent (*M. musculus*) | $Foxp3^{Cre}$ [$Foxp3^{tm4(YFP/icre)Ayr}$/J] | The Jackson Laboratory | Stock# 016959; RRID:IMSR_JAX:016959 | |
| Genetic reagent (*M. musculus*) | $Lrpprc^{f}$ [$Lrpprc^{tm1.1Lrsn}$/J] | Nils-Göran Larsson (***Ruzzenente et al., 2012***) | | |
| Genetic reagent (*M. musculus*) | $Miro1^{f}$ [B6(Cg)-$Rhot1^{tm2.1Jmsu}$/J] | The Jackson Laboratory | Stock# 031126; RRID:IMSR_JAX:031126 | |
| Genetic reagent (*M. musculus*) | $Pdgfa^{ex4COIN}$ [$Pdgfa^{ex4COIN}$] | ChristerBetsholtz (***Andrae et al., 2014***) | | |

*Appendix 1 Continued on next page*

*Appendix 1 Continued*

| Reagent type (species) or resource | Designation | Source or reference | Identifiers | Additional information |
|---|---|---|---|---|
| Genetic reagent (*M. musculus*) | *Pdgfra^{Cre}* [C57BL/6-Tg(Pdgfra-cre)1Clc/J] | The Jackson Laboratory | Stock# 013148; RRID:IMSR_JAX:013148 | |
| Genetic reagent (*M. musculus*) | *Rptor^f* [B6.Cg-*Rptor^{tm1.1Dmsa}*/J] | The Jackson Laboratory | Stock# 013188; RRID:IMSR_JAX:013188 | |
| Genetic reagent (*M. musculus*) | *ROSA26^{mTmG}* [*Gt(ROSA)26Sor^{tm4}(ACTB-tdTomato,-EGFP)^{Luo}*/J] | The Jackson Laboratory | Stock# 007576; RRID:IMSR_JAX:007576 | |
| Genetic reagent (*M. musculus*) | *Sox9^{Cre}* [Sox9^{tm3(Cre)Crm}] | Benoit de Crombrugghe (*Akiyama et al., 2005*) | | |
| Genetic reagent (*M. musculus*) | *Tfam^f* [B6.Cg-*Tfam^{tm1.1Ncdl}*/J] | The Jackson Laboratory | Stock# 026123; RRID:IMSR_JAX:026123 | |
| Biological sample (*H. sapiens*) | Human Emphysema/COPD patient tissue (deidentified) | Paul Wolters | | |
| Recombinant DNA reagent | Lentiviral vector backbone for *Pdgfa* cloning, modified from pSECC | This paper | | Refer to the 'Lentivirus production and transduction' |
| Recombinant DNA reagent | pSECC | *Sánchez-Rivera et al., 2014* | AddgenePlasmid# 60820 | |
| Recombinant DNA reagent | pMD2.G | A gift from Didier Trono | AddgenePlasmid# 12259 | |
| Recombinant DNA reagent | psPAX2 | A gift from Didier Trono | AddgenePlasmid# 12260 | |
| Software, algorithm | Prism | GraphPad | https://www.graphpad.com/ | |
| Software, algorithm | ImageJ | National Institutes of Health (NIH) (https://imagej.nih.gov/ij/) | RRID:SCR_003070 | |

