## [Editor Report]

This paper will be of interest to the large class of scientists interested in lung development and disease. It explores the under-investigated role of mitochondrial activity and subcellular distribution for alveolar formation, by using a variety of transgenic mouse models to delete two specific mitochondrial proteins. The data support a role for mitochondria distribution and function in postnatal lung development in the mice.

---

## [Decision Letter]

**Decision letter after peer review:**

Thank you for submitting your article "Acquisition of cellular properties during alveolar formation requires differential activity and distribution of mitochondria" for consideration by *eLife*. Your article has been reviewed by 3 peer reviewers, one of whom is a member of our Board of Reviewing Editors, and the evaluation has been overseen Edward Morrisey as the Senior Editor. The following individual involved in review of your submission has agreed to reveal their identity: David N Cornfield (Reviewer #3).

Essential revisions:

This manuscript explores the under-investigated role of mitochondrial activity and subcellular distribution for alveolar formation by using a variety of transgenic mouse models to delete two specific mitochondrial proteins. While largely descriptive, the data support a role for mitochondria distribution and function postnatal lung development in the mice. However, the experimental design and analysis as well as conclusions drawn within the manuscript raises several concerns, which are outlined below. The following points need to be addressed:

1) It remains unclear whether the effects of deleting the two mitochondrial molecules of interest may represent a general phenomenon as opposed to a specific effect in lung alveolarization. Please provide evidence of cell specific effect as well as evidence that the cells remain healthy in the face of the genetic manipulations. In detail, additional data/experiments to further characterize potential cell-specific effects/impairments upon genetic modulation including apoptosis/ viability and expression of cell-specific functional marker/transcriptional activity are required to significantly improve the manuscript. Further in vivo experiments using a non-structural (immune) cell driver could be helpful to understand whether this is a generalizable phenomenon or not.

2) Similarly, cell-specific effects can be further explored using vitro studies that need to be strengthened by gain- and loss-of-function experiments (potentially in human primary cells).

3) Additional parameters of lung histology, including radial alveolar count and septal thickness are required to evaluate the lung structure and function.

4) Currently insufficient data showing mitochondrial dysfunction upon knockdown, additional experiments to show that mitochondria are dysfunctional are needed.

5) Human disease link is currently vague and further clarification is needed on how these reported mitochondrial changes are unique to COPD/emphysema in comparison to other chronic lung diseases,

*Reviewer #1 (Recommendations for the authors):*

It would significantly enhance the manuscript if the mechanisms suggested based on the in vivo mouse data would have been expanded on, in particular with respect to a potential human disease. For example, using primary fibroblasts and performing additional gain and loss of function studies on Miro1 and *Tfam* and investigation of the cellular (mitochondrial) signaling as well as functional consequence, would be helpful.

With respect to COPD/emphysema, further clarification is needed on how these reported mitochondrial changes are unique to COPD/emphysema in comparison to other chronic lung diseases, for which mitochondrial dysfunction has been reported. Given the reported relevance for postnatal development, diseases such as bronchopulmonary dysplasia are important to discuss.

*Reviewer #2 (Recommendations for the authors):*

This manuscript provides a novel insight that mitochondria bring impact on alveolar formation. However, there are following questions and concerns to be addressed:

1. The current conclusion on defective alveologenesis as a consequence of mitochondrial dysfunction is questionable due to insufficient data showing mitochondrial dysfunction. Confirmation studies on mitochondrial function (eg. Oroboros assay, ATP assay) are required. Furthermore, mitochondrial network analysis is essential for confirming the alteration of mitochondrial activity and dynamic distribution.

2. Since mouse line of *Tfam^f/f^; Pdgfa^ex4COIN/+^; Sox9^Cre/+^* cannot properly indicates Pdgfa expression, please use both mouse line of *Pdgfa^ex4COIN/+^; Sox9^Cre/+^* and mouse line of *Tfam^f/f^; Sox9^Cre/+^* as control groups or measure the PDGF expression via other techniques.

3. It is interesting that COPD/emphysema patients show decreased mitochondrial copy number and *TFAM* protein levels. However, it would be better if authors further analyze these alterations upon cell types. Moreover, please discuss how mitochondrial alterations and disrupted secondary septa formation link to COPD/emphysema pathogenesis.

4. Lentiviral transduction efficiency is critical for the data quality of PDGF levels. Please provide it.

5. Please provide Western blotting images used for quantification in Figure 7E, Figure 8C and Figure 8E.

*Reviewer #3 (Recommendations for the authors):*

Overall, there is a great deal of merit to the study. Demonstration that the effects are cell specific, and not the result of a generalized phenomena would be helpful. To this end, demonstration that the cell with altered mitochondria are otherwise robust would benefit the manuscript. In addition, demonstration that some cellular constituent of the alveolar still carries out its biological role in the face of compromised mitochondrial function or number, and that alveolarization is unperturbed would be important. In the current form, it is difficult to distinguish a cell-specific effect. Given these limitations, a more focused Discussion, clearly addressing these limitations would be helpful.

Further, the histologic analysis of the lung should be more comprehensive and include radial alveolar counts as well as septal thickness. Mean linear intercept addresses the size of the alveolus, but not number, even though these parameters often track one another. More well curated markers of different cell types to highlight the specificity of the drivers used would be helpful.

1. In the Results section, Figure 1, *Sox9* was used as a specific marker for lung epithelial cells. Is it specific? If so, which of the epithelial are *Sox9* expressors, alveolar type 1, type 2, both?

2. In the same section, the authors comment that myofibroblasts are marked by PDGFRA, PDGF receptor and smooth muscle actin. What is the distinction between each of the first two markers?

3. Relative to the mitochondria, cells require energy to function, so the need to mitochondria is anticipated. Are the authors suggesting a cell-specific effect? Further, is the inhomogeneous distribution of mitochondria within the cells of interest in contrast with homogeneous distribution in cells less dependent upon the mitochondria?

4. What were the alveolar defects aside an increase in mean linear intercept? For example, was septal thickness affected? Alternatively, was radial alveolar count affected?

5. On page7, the authors state "Moreover, histological analysis revealed no difference between control and mutant lungs prior to P5…" there certainly appears to be histologic differences in terms of septal thickness. Given the differences in histologic appearance it seems that lung histology, perhaps mesenchymal histology, was affected by inactivating *Tfam* in *SOX9*^+^ cells

6. While *Miro1^f/f^; Sox9Cre/+* mice had alterations in the distribution of mitochondria, were absolute numbers of mitochondria decreased as well? Again, lung histology does appear different at P5, even though not in terms of mean linear intercept. Overall, the manuscript would benefit from additional parameters of lung histology, including radial alveolar count and septal thickness.

7. Relative to the mesenchymal studies using the Pdgfra driver, which cell or cellular constituents in the mesenchyme are being affected? The mesenchyme includes multiple, distinct cell types including transcriptomically distinct fibroblasts, myofibroblasts, and airway smooth muscle cells.

8. In the myofibroblast migration assays, was there a rescue component to the experiment? Alternatively, was loss of migratory capacity interrogated other than through decreases in mitochondrial number, function? IN general, the in vitro studies could be strengthened by gain- and loss-of-function components.

9. Relative to the lung histology, it is important to standardize tissue acquisition, especially inflation of the lung. Detail surrounding how this was done would be helpful.

10. In Figure 8, the results are normalized to the controls. If the absolute data are presented, as opposed to % of control, are the differences significant?

[Editors' note: further revisions were suggested prior to acceptance, as described below.]

Thank you for resubmitting your work entitled "Acquisition of cellular properties during alveolar formation requires differential activity and distribution of mitochondria" for further consideration by *eLife*. Your revised article has been evaluated by Edward Morrisey (Senior Editor) and a Reviewing Editor.

The manuscript has been improved but there are some remaining issues that need to be addressed, as outlined below. Specifically, the concerns related to cell specificity has not been addressed and the fact that no direct evidence is provided to show that modulation of mitochondrial function in other cells will not affect on lung structure/alveolarization. This is a critical point. Additional concerns about the clarity and interpretation will further need to be addressed.

*Reviewer #2 (Recommendations for the authors):*

The authors have answered most of the questions raised in the previous reviews and revised the manuscript.

*Reviewer #3 (Recommendations for the authors):*

The rebuttal response addresses components of the concerns of the reviewers. However, the most fundamental concern remains unaddressed. Specifically, is there a cell intrinsic to lung structure wherein mitochondrial function is diminished that has no effect on secondary septation. In the reply, the authors point out that the saccular stage might be unaffected (Figure 3), but not evidence that alveolarization is preserved despite genetic deletion. It seems that the lines of evidence presented are each indirect. The concern about cell specificity was the central concern expressed by 2 of 3 reviewers. It seems that such data is critical in order to defend the claims of cell specificity.

The addition of data demonstrating mitochondrial dysfunction is helpful

The additional lung histology is helpful. Do the differences in septal thicknesses merit comment?

Why is the observation that saccular formation is unperturbed while alveolarization is compromised a reflection of differential energy requirements as opposed to involvement of different cells at specific stages of lung development?

The authors comment about BPD in the abstract. Might the cell specific compromise in mitochondrial function have implications for BPD?

[Editors' note: further revisions were suggested prior to acceptance, as described below.]

Thank you for submitting your article "Acquisition of cellular properties during alveolar formation requires differential activity and distribution of mitochondria" for consideration by *eLife*. Your article has been reviewed by 2 peer reviewers, and the evaluation has been overseen by a Reviewing Editor and Edward Morrisey as the Senior Editor. The reviewers have opted to remain anonymous.

Essential revisions:

The manuscript is much improved by including the additional data. Reviewer 3 raises important concerns that we ask you to address in the manuscript text throughout as well as by specifically revising your conclusions and clearly stating the limitations regarding epithelial cell-specificity for the effects observed. No additional experiments are required. This paper will be of interest to the large class of scientists interested in lung development and disease. It explores the under-investigated role of mitochondrial activity and subcellular distribution for alveolar formation, by using a variety of transgenic mouse models to delete two specific mitochondrial proteins. While largely descriptive, the data support a role for mitochondria distribution and function in postnatal lung development in the mice.

*Reviewer #3 (Recommendations for the authors):*

The present revision includes a number of well performed and carefully considered experiments. The authors have responded to the concerns raised in each of the prior reviews. However, the primary concern persists. Specifically, the authors argue that in *Sox9* expressing epithelial cells mitochondrial distribution and activity have specific effects on secondary septation of the lung. Doubtless, the manuscript includes a rich set of experiments. The authors definitively demonstrate that compromising mitochondrial activity and distribution in *Sox9* expressing epithelial cells decreases secondary septation. However, it does not seem to this Reviewer that the authors can make the claims of specificity outlined in the manuscript. To make such a claim, it would be important, as stated in the prior reviews, by more than one Reviewer, to demonstrate that loss of mitochondrial activity and altered distribution in another cell type in the lung has no effect on alveolarization. The demonstration that loss of mitochondrial activity non-structural immune cells is helpful, these T-cells compromise a very small percentage of the overall immune population and remain naive through ~ P21. Moreover, there are no reports of a lung phenotype in mice wherein T-cells are deleted; if deleting cells has no effect on alveolarization, then certainly decreasing mitochondrial activity would have no effect. A non-structural immune cell, such as macrophages would be far more meaningful proof of concept.

Relative to claims of specificity regarding use of a Pdgra promoter to effect myofibroblasts problematic. As the authors point out, fibroblasts, adventitial fibroblasts, and adventitial fibroblasts (transcriptionally distinct subpopulations) all express Pdgfra and thus the knockout is certainly effecting cells other than the myofibroblasts.

What is the basis for the contention that septal thickness is a more sensitive indicator of secondary septation? Thinning of the septae is a component of lung maturation, but not specifically secondary septation.

Transcriptomically distinct cell subtypes emerge during the transition from saccular to alveolar stages. This has been shown in multiple papers. Thus, it remains unclear just why the authors insist that the absence of effects during the saccular stage proves differential energy requirements in the same cells as opposed to involvement of alternative cells.

Though not appreciated by the authors, radial alveolar count is distinct from mean linear intercept and represents a more direct method to secondary septae, the outcome measure of primary concern in the present manuscript.

In Figure 6, line 940, how do the authors know the cells were myofibroblasts?

Finally, in Figure 8, it appears that panels b,c, and e are normalized so that the controls = 1. Are findings still significant if the absolute values are compared?

---

## [Author Response]

Essential revisions:This manuscript explores the under-investigated role of mitochondrial activity and subcellular distribution for alveolar formation by using a variety of transgenic mouse models to delete two specific mitochondrial proteins. While largely descriptive, the data support a role for mitochondria distribution and function postnatal lung development in the mice. However, the experimental design and analysis as well as conclusions drawn within the manuscript raises several concerns, which are outlined below. The following points need to be addressed:1) It remains unclear whether the effects of deleting the two mitochondrial molecules of interest may represent a general phenomenon as opposed to a specific effect in lung alveolarization. Please provide evidence of cell specific effect as well as evidence that the cells remain healthy in the face of the genetic manipulations. In detail, additional data/experiments to further characterize potential cell-specific effects/impairments upon genetic modulation including apoptosis/ viability and expression of cell-specific functional marker/transcriptional activity are required to significantly improve the manuscript. Further in vivo experiments using a non-structural (immune) cell driver could be helpful to understand whether this is a generalizable phenomenon or not.

Multiple pieces of data in the original submission suggest that a reduction in mitochondrial function does not exert a uniform effect on various cell types or tissues. While ATP production by mitochondria is essential for proper functioning of all cell types and tissues, our results suggest that their requirement of ATP levels varies, depending on the cell type and the developmental stage. Employment of appropriate mouse Cre lines is critical to reducing mitochondrial function in the targeted cell types and stages.

1) Removal of murine *Tfam* by *Sox9-Cre* in the lung epithelium or *Miro1* (*Rhot1*) by *Pdgfra-Cre* in the lung mesenchyme did not affect the number and distribution of club cells, ciliated cells, alveolar type I cells and type II cells (Figure 3—figure supplement 2 and Figure 5—figure supplement 1).

2) Loss of *Tfam* or *Miro1* in the lung epithelium or mesenchyme did not perturb saccule formation (Figure 3A, 3G, 5A, 5G).

3) Myofibroblast proliferation was unaltered for a certain period of time following *Tfam* or *Miro1* removal in the lung epithelium or mesenchyme (Figure 4D, 6B, 6F).

4) The expression of *Pdgfa* (mainly in alveolar epithelial cells) and PDGFRA (in alveolar fibroblasts/myofibroblasts) was largely unchanged in the absence of epithelial *Tfam* or *Miro1* or mesenchymal *Miro1* (Figure 4E, 4J, 6G).

5) The rate of apoptosis was similar between control and *Tfam*- or *Miro1*-deficient lungs (Figure 3—figure supplement 3 and Figure 5—figure supplement 2).

Moreover, we have produced *Tfam^f/f^; Foxp3^Cre/+^* mice in which *Tfam* was inactivated by *Foxp3-Cre* in T cells. These mice did not exhibit alveolar defects (Figure 3—figure supplement 5), further supporting differential requirement of ATP levels. This is also consistent with data in the published literature, where immune defects but not alveolar defects in *Tfam^f/f^; Foxp3^Cre/+^* mice were noted.

2) Similarly, cell-specific effects can be further explored using vitro studies that need to be strengthened by gain- and loss-of-function experiments (potentially in human primary cells).

As suggested by the reviewers, we have conducted gain- and loss-of-function studies on *TFAM* and *MIRO1* in both human lung epithelial cells and human lung fibroblasts. Assays were focused on PDGF secretion from epithelial cells and cell migration of fibroblasts. Results from the experiments using human lung cells (Figure 8F, 8G, Figure 8—figure supplement 3) affirmed the findings obtained in mouse cells and mouse lungs.

3) Additional parameters of lung histology, including radial alveolar count and septal thickness are required to evaluate the lung structure and function.

We have measured both the radial alveolar account and septal thickness as suggested by the reviewers. Since the methodology of measuring radial alveolar count is similar to that of mean linear intercept, the result of radial alveolar count parallels that of mean linear intercept as expected. The septal thickness was reduced in *Tfam*- and *Miro1*-deficient lungs as shown in Figure 3C, 3I and Figure 5C, 5I in the revised manuscript.

4) Currently insufficient data showing mitochondrial dysfunction upon knockdown, additional experiments to show that mitochondria are dysfunctional are needed.

In the revision, we have provided additional data on mitochondrial dysfunction in the mutant lungs. They include assays that determine changes in the regulators of mitochondrial fusion and fission (Figure 3—figure supplement 4), mitochondrial network (Figure 6—figure supplement 1), enzymatic activities of mitochondrial complexes 1 and 4 (Figure 3E, 3K, 5E, 5K), and ATP production (Figure 3F, 3L, 5F, 5L).

5) Human disease link is currently vague and further clarification is needed on how these reported mitochondrial changes are unique to COPD/emphysema in comparison to other chronic lung diseases,

While mitochondrial dysfunction is associated with human COPD/emphysema, it is difficult to establish a causal relationship, a general problem in studying human diseases. Nevertheless, we have speculated in the revised manuscript how mitochondrial dysfunction in lung epithelial cells may affect epithelial function and subsequently mesenchymal cell properties. Disturbance in epithelial and mesenchymal cells could contribute to the pathogenesis of COPD/emphysema.

Reviewer #1 (Recommendations for the authors):It would significantly enhance the manuscript if the mechanisms suggested based on the in vivo mouse data would have been expanded on, in particular with respect to a potential human disease. For example, using primary fibroblasts and performing additional gain and loss of function studies on Miro1 and *Tfam* and investigation of the cellular (mitochondrial) signaling as well as functional consequence, would be helpful.

As suggested by the reviewer, we have conducted gain- and loss-of-function studies on *TFAM* and *MIRO1* (*RHOT1*) in both human lung epithelial cells and human lung fibroblasts. Assays were focused on PDGF secretion from epithelial cells and cell migration of fibroblasts. Results from the experiments using human lung cells (Figure 8F, 8G, Figure 8—figure supplement 3) affirmed the findings obtained in mouse cells and mouse lungs.

With respect to COPD/emphysema, further clarification is needed on how these reported mitochondrial changes are unique to COPD/emphysema in comparison to other chronic lung diseases, for which mitochondrial dysfunction has been reported. Given the reported relevance for postnatal development, diseases such as bronchopulmonary dysplasia are important to discuss.

While mitochondrial dysfunction is associated with human COPD/emphysema, it is difficult to establish a causal relationship, a general problem in studying human diseases. Nevertheless, we have speculated in the revised manuscript how mitochondrial dysfunction in lung epithelial cells may affect epithelial function and subsequently mesenchymal cell properties. Disturbance in epithelial and mesenchymal cells could contribute to the pathogenesis of COPD/emphysema.

Reviewer #2 (Recommendations for the authors):This manuscript provides a novel insight that mitochondria bring impact on alveolar formation. However, there are following questions and concerns to be addressed:1. The current conclusion on defective alveologenesis as a consequence of mitochondrial dysfunction is questionable due to insufficient data showing mitochondrial dysfunction. Confirmation studies on mitochondrial function (eg. Oroboros assay, ATP assay) are required. Furthermore, mitochondrial network analysis is essential for confirming the alteration of mitochondrial activity and dynamic distribution.

As suggested by the reviewer, in the revision, we have provided additional data on mitochondrial dysfunction in control and *Tfam*- and *Miro1* (*Rhot1*)-deficient lungs using select mouse Cre lines. They include assays that determine the enzymatic activities of mitochondrial complexes 1 and 4 (Figure 3E, 3K, 5E, 5K) and ATP production (Figure 3F, 3L, 5F, 5L) in control and mutant lungs. We have also analyzed the mitochondrial network in control and mutant cells (Figure 6—figure supplement 1.

2. Since mouse line of *Tfam^f/f^; Pdgfa^ex4COIN/+^; Sox9^Cre/+^* cannot properly indicates Pdgfa expression, please use both mouse line of *Pdgfa^ex4COIN/+^; Sox9^Cre/+^* and mouse line of *Tfam^f/f^; Sox9^Cre/+^* as control groups or measure the PDGF expression via other techniques.

In Figure 4C, the complete genotype of the control mouse is *Sox9^Cre/+^; Pdgfa^ex4COIN/+^*, which has been clarified. We have also included *Pdgfa^ex4COIN/+^* and *Tfam^f/f^; Sox9^Cre/+^* lungs as controls (Figure 4—figure supplement 1). The transcript levels of *Pdgfa* is shown in Figure 4E as an alternative means to measure PDGFA expression.

3. It is interesting that COPD/emphysema patients show decreased mitochondrial copy number and *TFAM* protein levels. However, it would be better if authors further analyze these alterations upon cell types. Moreover, please discuss how mitochondrial alterations and disrupted secondary septa formation link to COPD/emphysema pathogenesis.

We were unable to secure a sufficient number of cell lines derived from COPD/emphysema patients for analysis of alterations in different cell types. As an alternative approach, we employed human lung epithelial cells and lung fibroblasts. Gain- and loss-of-function studies on *TFAM* and *MIRO1* were conducted in human lung epithelial cells and human lung fibroblasts. Assays were focused on PDGF secretion from epithelial cells and cell migration of fibroblasts. Results from the experiments using human lung cells (Figure 8F, 8G, Figure 8—figure supplement 3) affirmed the findings obtained in mouse cells and mouse lungs.

We have added discussions on the relationship between mitochondrial alteration, disrupted secondary septation and COPD/emphysema pathogenesis in the revised manuscript.

4. Lentiviral transduction efficiency is critical for the data quality of PDGF levels. Please provide it.

We have added data on lentiviral transduction efficiency in Figure 4—figure supplement 1. The efficiency of lentiviral transduction was not affected by the loss of *Tfam* or *Miro1*.

5. Please provide Western blotting images used for quantification in Figure 7E, Figure 8C and Figure 8E.

We have provided Westerns blotting images in Figure 7—figure supplement 1 and Figure 8—figure supplement 2.

Reviewer #3 (Recommendations for the authors):Overall, there is a great deal of merit to the study. Demonstration that the effects are cell specific, and not the result of a generalized phenomena would be helpful. To this end, demonstration that the cell with altered mitochondria are otherwise robust would benefit the manuscript. In addition, demonstration that some cellular constituent of the alveolar still carries out its biological role in the face of compromised mitochondrial function or number, and that alveolarization is unperturbed would be important. In the current form, it is difficult to distinguish a cell-specific effect. Given these limitations, a more focused Discussion, clearly addressing these limitations would be helpful.Further, the histologic analysis of the lung should be more comprehensive and include radial alveolar counts as well as septal thickness. Mean linear intercept addresses the size of the alveolus, but not number, even though these parameters often track one another. More well curated markers of different cell types to highlight the specificity of the drivers used would be helpful.1. In the Results section, Figure 1, Sox9 was used as a specific marker for lung epithelial cells. Is it specific? If so, which of the epithelial are Sox9 expressors, alveolar type 1, type 2, both?

It is known that *Sox9-Cre*, *Shh-Cre*, *Nkx2.1-Cre* and *SPC-Cre* can label most lung epithelial cells (but not lung mesenchymal cells) through a reporter such as *ROSA26^mTmG^*. Thus, alveolar type I and type II cells would be labeled by GFP in *Sox9^Cre/+^; ROSA26^mTmG/+^* lungs. This point has been described in our earlier publications (*e.g.*, Zhang et al. *eLife*, 2020) and those from other groups.

2. In the same section, the authors comment that myofibroblasts are marked by PDGFRA, PDGF receptor and smooth muscle actin. What is the distinction between each of the first two markers?

In alveolar myofibroblasts, smooth muscle actin (SMA or ACTA2) can only be detected by immunostaining from P3 to P12. After P10, SMA expression sharply decreases in alveolar myofibroblasts, but is retained in airway smooth muscle or vascular smooth muscle cells. Unlike SMA, PDGFRA expression persists in alveolar myofibroblasts.

3. Relative to the mitochondria, cells require energy to function, so the need to mitochondria is anticipated. Are the authors suggesting a cell-specific effect? Further, is the inhomogeneous distribution of mitochondria within the cells of interest in contrast with homogeneous distribution in cells less dependent upon the mitochondria?

*Tfam* removal reduces but does not completely obliterate ATP production. ~30–50% reduction in ATP levels in the whole lungs was observed (Figure 3F, 3L and Figure 5F, 5L). This is expected since all cells require ATP for proper functioning. We anticipate that a partial reduction in ATP levels will preferentially affect cellular processes with a higher energy demand. We showed that loss of epithelial or mesenchymal *Tfam* resulted in alveolar defects but not branching morphogenesis or saccule formation. This implies that secondary septation requires higher ATP levels compared to other cellular processes. We speculate that ligand secretion from the lung epithelium and myofibroblast contraction and migration have a greater demand for ATP. Consistent with this notion, mitochondria in alveolar epithelial cells accumulate in regions close to the trans Golgi network, while mitochondria in alveolar myofibroblasts concentrate in areas near the smooth muscle actin. Inhomogeneous distribution of mitochondria in a given cell type may be an indicator of higher energy demand for that cell type.

4. What were the alveolar defects aside an increase in mean linear intercept? For example, was septal thickness affected? Alternatively, was radial alveolar count affected?

In *Tfam* or *Miro1* mutant mice using select Cre lines, the proliferation rate of alveolar myofibroblasts was decreased. This would reduce mesenchymal cell number in the primary septa. Thus, the septal thickness could serve to characterize the alveolar defect, in addition to the mean linear intercept. The septal thickness was reduced in *Tfam*- and *Miro1* (*Rhot1*)-deficient lungs as shown in Figure 3C, 3I and Figure 5C, 5I in the revised manuscript.

Since the methodology of measuring radial alveolar count is similar to that of mean linear intercept, the result of radial alveolar count parallels that of mean linear intercept as expected.

5. On page7, the authors state "Moreover, histological analysis revealed no difference between control and mutant lungs prior to P5…" there certainly appears to be histologic differences in terms of septal thickness. Given the differences in histologic appearance it seems that lung histology, perhaps mesenchymal histology, was affected by inactivating *Tfam* in SOX9^+^ cells

We stated that there is no apparent difference between control and mutant (*Tfam^f/f^; Sox9^Cre/+^*) lungs prior to P5. At P5, lung phenotypes in certain regions of mutant lungs could be discerned as shown in Figure 3A. Since the lung phenotype in the mutants at P5 was relatively mild and non-uniform, MLI, an indicator of the average airspace, was similar between control and mutant lungs. Nevertheless, measurement of septal thickness at P5 did reveal a difference between control and mutant lungs as shown in Figure 3C.

6. While *Miro1^f/f^; Sox9^Cre/+^* mice had alterations in the distribution of mitochondria, were absolute numbers of mitochondria decreased as well? Again, lung histology does appear different at P5, even though not in terms of mean linear intercept. Overall, the manuscript would benefit from additional parameters of lung histology, including radial alveolar count and septal thickness.

We did not observe an alteration in the mtDNA/nDNA ratio in *Miro1^f/f^; Sox9^Cre/+^* lungs. This suggests that the absolute number of mitochondria was unaffected in the absence of epithelial *Miro1*. This piece of data is shown in Figure 3J of the revised manuscript.

We have measured both the radial alveolar account and septal thickness as suggested by the reviewer. Since the methodology of measuring radial alveolar count is similar to that of mean linear intercept, the result of radial alveolar count parallels that of mean linear intercept as expected. The septal thickness was reduced in *Tfam*- and *Miro1*-deficient lungs as shown in Figure 3C, 3I and Figure 5C, 5I in the revised manuscript.

7. Relative to the mesenchymal studies using the Pdgfra driver, which cell or cellular constituents in the mesenchyme are being affected? The mesenchyme includes multiple, distinct cell types including transcriptomically distinct fibroblasts, myofibroblasts, and airway smooth muscle cells.

In *Tfam^f/f^; Pdgfra^Cre/+^* and *Miro1^f/f^; Pdgfra^Cre/+^* lungs, we did not observe an apparent defect in airway smooth muscle cells and vascular smooth muscle cells. No difference in differentiation, migration, structure and marker expression could be discerned between control and mutant lungs. By contrast, alveolar fibroblasts/myofibroblasts were the main cell type affected in the mutant lungs.

8. In the myofibroblast migration assays, was there a rescue component to the experiment? Alternatively, was loss of migratory capacity interrogated other than through decreases in mitochondrial number, function? IN general, the in vitro studies could be strengthened by gain- and loss-of-function components.

We have added rescue data for the myofibroblast migration assay in Figure 6—figure supplement 2.

We have also included gain- and loss-of-function studies on *TFAM* and MIRO1 in the revised manuscript. These experiments were conducted in human lung epithelial and fibroblast cell lines and presented in Figure 8F, 8G and Figure 8—figure supplement 3.

9. Relative to the lung histology, it is important to standardize tissue acquisition, especially inflation of the lung. Detail surrounding how this was done would be helpful.

In this study, acquisition of all tissues (including the lungs) followed a standardized procedure as detailed in Materials and methods. Of note, we did not inflate the lungs to avoid mechanical distortion of the lung structure.

10. In Figure 8, the results are normalized to the controls. If the absolute data are presented, as opposed to % of control, are the differences significant?

When the absolute data, instead of % of control, were used for statistical analysis, the same p value was obtained.

[Editors' note: further revisions were suggested prior to acceptance, as described below.]

The manuscript has been improved but there are some remaining issues that need to be addressed, as outlined below. Specifically, the concerns related to cell specificity has not been addressed and the fact that no direct evidence is provided to show that modulation of mitochondrial function in other cells will not affect on lung structure/alveolarization. This is a critical point. Additional concerns about the clarity and interpretation will further need to be addressed.Reviewer #3 (Recommendations for the authors):The rebuttal response addresses components of the concerns of the reviewers. However, the most fundamental concern remains unaddressed. Specifically, is there a cell intrinsic to lung structure wherein mitochondrial function is diminished that has no effect on secondary septation. In the reply, the authors point out that the saccular stage might be unaffected (Figure 3), but not evidence that alveolarization is preserved despite genetic deletion. It seems that the lines of evidence presented are each indirect. The concern about cell specificity was the central concern expressed by 2 of 3 reviewers. It seems that such data is critical in order to defend the claims of cell specificity.

In Essential Revisions, Q1 of the first round of review, it was suggested that “Further in vivo experiments using a non-structural (immune) cell driver could be helpful to understand whether this is a generalizable phenomenon or not.” We have conducted this experiment and produced *Tfam^f/f^; Foxp3^Cre/+^* mice in which *Tfam* was inactivated by *Foxp3-Cre* in T cells. These mice did not exhibit alveolar defects (Figure 3—figure supplement 5). Since then, we have also generated *Tfam^f/f^; Cd4^Cre/+^* mice to remove *Tfam* in T cells using *Cd4-Cre*. These mice also did not display alveolar defects (revised Figure 3—figure supplement 5).

Since mitochondrial function is essential for all cells, we anticipate that reduced mitochondrial function in any structural component of the secondary septa would lead to alveolar defects. We would like to point out that our study provides key insights into how mitochondrial activity and distribution control certain cellular processes in different components of the secondary septa that are critical for secondary septation. As mentioned in the previous response, not all aspects of cellular processes are affected in *Tfam*-deficient cells. For instance, expression of *Pdgfa* and PDGFRA, proliferation and apoptosis, and saccule formation (also see below) do not seem to be affected.

In response to the reviewer’s comment, we have revised the text to indicate the susceptibility of structural vs. non-structural components of the secondary septa to reduced mitochondrial function.

The addition of data demonstrating mitochondrial dysfunction is helpful

We appreciate the reviewer’s comment.

The additional lung histology is helpful. Do the differences in septal thicknesses merit comment?

In the first revision, we stated that “…various postnatal stages revealed defects in secondary septa formation with an increased MLI (Figure 3A, 3B) and reduced septal thickness (Figure 3C)“. In response to the reviewer’s comment, we have now added “Septal thickness is more sensitive than MLI in detecting defects in secondary septation”.

Why is the observation that saccular formation is unperturbed while alveolarization is compromised a reflection of differential energy requirements as opposed to involvement of different cells at specific stages of lung development?

The main constituents of both saccules and alveoli are alveolar type I and type II cells and alveolar fibroblasts/myofibroblasts. Alveolar formation occurs within saccules as a continuous developmental process. As such, it appears that the simplest explanation is that the same cells have distinct requirements at different developmental time points.

The authors comment about BPD in the abstract. Might the cell specific compromise in mitochondrial function have implications for BPD?

BPD is often caused by high oxygen levels. Given the central role of mitochondria in oxidative phosphorylation and ATP production, it is possible that cell-specific compromise in mitochondrial function could be related to BPD. We do not have any evidence to show such a connection. This would require future investigations.

[Editors' note: further revisions were suggested prior to acceptance, as described below.]

Essential revisions:The manuscript is much improved by including the additional data. Reviewer 3 raises important concerns that we ask you to address in the manuscript text throughout as well as by specifically revising your conclusions and clearly stating the limitations regarding epithelial cell-specificity for the effects observed. No additional experiments are required.

We have modified the conclusions, stated the limitation of our approaches throughout the revised manuscript. In particular, we directly addressed the issue of cell-specificity for the effects observed. We have also added new data in which we showed that removal of *Tfam* in alveolar macrophages did not lead to alveolar defects. This is similar to the finding that *Tfam* inactivation in T cells has no apparent effects on alveologenesis.

Reviewer #3 (Recommendations for the authors):The present revision includes a number of well performed and carefully considered experiments. The authors have responded to the concerns raised in each of the prior reviews. However, the primary concern persists. Specifically, the authors argue that in Sox9 expressing epithelial cells mitochondrial distribution and activity have specific effects on secondary septation of the lung. Doubtless, the manuscript includes a rich set of experiments. The authors definitively demonstrate that compromising mitochondrial activity and distribution in Sox9 expressing epithelial cells decreases secondary septation. However, it does not seem to this Reviewer that the authors can make the claims of specificity outlined in the manuscript. To make such a claim, it would be important, as stated in the prior reviews, by more than one Reviewer, to demonstrate that loss of mitochondrial activity and altered distribution in another cell type in the lung has no effect on alveolarization. The demonstration that loss of mitochondrial activity non-structural immune cells is helpful, these T-cells compromise a very small percentage of the overall immune population and remain naive through ~ P21. Moreover, there are no reports of a lung phenotype in mice wherein T-cells are deleted; if deleting cells has no effect on alveolarization, then certainly decreasing mitochondrial activity would have no effect. A non-structural immune cell, such as macrophages would be far more meaningful proof of concept.

In the previous revision, we showed that mouse lungs with *Tfam* inactivated in T cells using *Foxp3-Cre* or *CD4-Cre* do not show alveolar defects (Figure 3—figure supplement 5). Loss of *Tfam* by *Foxp3-Cre* has been previously reported (Fu et al., *Cell Rep* 2019 PMID 31269437). These animals displayed severe inflammation but no alveolar defects.

Of note, ample evidence suggests that neonatal T cells are not immature or defective versions of adult cells (Rudd *Annu Rev Immunol* 2020 PMID 31926469). They are well adapted to provide fast-acting immune responses.

The reviewer felt that deletion of *Tfam* in macrophages would be far more meaningful. This study has been published in which *Tfam* removal in alveolar macrophages by *Cd11c-Cre* does not lead to alveolar defects (Gao et al., *J Immunol* 2022 PMID: 35165165). To further substantiate this finding, we have conditionally inactivated *Tfam* using *Cx3cr1-CreER*. We generated *Tfam^f/f^; Cx3cr1^CreER/+^* mice and administered tamoxifen to neonatal mice. These animals also did not display alveolar defects (Figure 3—figure supplement 5). Taken together, while these studies are not exhaustive, they provide strong support to our model in which lung epithelial and mesenchymal cells are more sensitive to perturbation of mitochondrial dysfunction than other cell types during alveologenesis.

We also would like to emphasize that saccule formation is not disrupted in *Tfam^f/f^; Sox9^Cre/+^* lungs. Thus, loss of *Tfam* or mitochondrial function does not have a universal effect on the developmental processes, which is a main point of our manuscript.

Relative to claims of specificity regarding use of a Pdgra promoter to effect myofibroblasts problematic. As the authors point out, fibroblasts, adventitial fibroblasts, and adventitial fibroblasts (transcriptionally distinct subpopulations) all express Pdgfra and thus the knockout is certainly effecting cells other than the myofibroblasts.

We acknowledged that *Pdgfra-Cre* labels many subsets of fibroblasts. We have adopted the more precise description of PDGFRA^+^ cells or fibroblasts/myofibroblasts in the revised manuscript. Nevertheless, it is worth mentioning that ~95% of lineaged cells (PDGFRA^+^) in *Pdgfra^rtTA^; tetO^Cre^* mice are myofibroblasts (Li et al., *ELife* 2018 PMID: 30178747).

What is the basis for the contention that septal thickness is a more sensitive indicator of secondary septation? Thinning of the septae is a component of lung maturation, but not specifically secondary septation.

We measured the primary septal thickness and MLI. Thinning of the primary septae appeared earlier than changes in MLI in our analysis. We surmise that thinning of the primary septae is caused by reduced myofibroblast number and contraction/migration, all of which will subsequently lead to defective secondary septation.

Transcriptomically distinct cell subtypes emerge during the transition from saccular to alveolar stages. This has been shown in multiple papers. Thus, it remains unclear just why the authors insist that the absence of effects during the saccular stage proves differential energy requirements in the same cells as opposed to involvement of alternative cells.

While there is no evidence that a new epithelial cell type emerges from saccular to alveolar stages, the transcriptome of any cell type changes as development proceeds. The simplest model is that the same cell types with an altered transcriptome (cell subtypes by the reviewer) display different sensitivity to mitochondrial perturbation (and energy expenditure) as lung development proceeds from saccular to alveolar stages. This is based on the assumption that the subcellular events (*e.g.*, ligand secretion and cellular contraction) that drive a cellular process (*e.g.*, sacculation and secondary septation) are the main consumer of energy for a given cell subtype. However, it is also possible that the subcellular events that execute a given cellular process may not be the main consumer of energy. In this scenario, one would conclude that the phenotypic defects of a given cellular process due to compromised mitochondrial activity are indicative of the sensitivity of the subcellular events to mitochondrial activity. We have clarified these points in the revised manuscript.

Though not appreciated by the authors, radial alveolar count is distinct from mean linear intercept and represents a more direct method to secondary septae, the outcome measure of primary concern in the present manuscript.

A large fraction of the lungs analyzed for alveolar defects in this study were from postnatal day 3 –5. Since no alveoli are formed at this stage, the method of radial alveolar count (RAC) is not applicable. The values of RAC would be zero by definition despite the presence of differences in the primary septal thickness and MLI between control and mutant lungs.

In Figure 6, line 940, how do the authors know the cells were myofibroblasts?

The cells in Figure 6A and 6E expressed SMA and are alveolar myofibroblasts. The cells in Figure D and 6H contained a mixture of fibroblasts and myofibroblasts.

Finally, in Figure 8, it appears that panels b,c, and e are normalized so that the controls = 1. Are findings still significant if the absolute values are compared?

When the absolute values in panel *B* are compared, the same p value is obtained.

For panels *C* and *E*, variabilities in loading and exposure for each scan of Western blotting necessitate normalization before data pooling for statistical analysis. This is a widely accepted practice.